# Wnt11 directs nephron progenitor polarity and motile behavior ultimately determining nephron endowment

Lori L O'Brien[1†]*, Alexander N Combes[2,3,4], Kieran M Short[5,6], Nils O Lindström[1], Peter H Whitney[1], Luise A Cullen-McEwen[5], Adler Ju[2], Ahmed Abdelhalim[1], Odyssé Michos[1], John F Bertram[5], Ian M Smyth[5,6], Melissa H Little[2,3,4,7], Andrew P McMahon[1]*

[1]Department of Stem Cell Biology and Regenerative Medicine, Eli and Edythe Broad CIRM Center for Regenerative Medicine and Stem Cell Research, Keck School of Medicine, University of Southern California, Los Angeles, United States; [2]Institute for Molecular Bioscience, The University of Queensland, Brisbane, Australia; [3]Department of Anatomy and Neuroscience, The University of Melbourne, Melbourne, Australia; [4]Murdoch Children's Research Institute, Royal Children's Hospital, Melbourne, Australia; [5]Department of Anatomy and Neuroscience, Monash Biomedicine Discovery Institute, Monash University, Melbourne, Australia; [6]Development and Stem Cells Program, Monash Biomedicine Discovery Institute, Monash University, Melbourne, Australia; [7]Department of Pediatrics, University of Melbourne, Parkville, Australia

*For correspondence:
lori_obrien@med.unc.edu (LLO'B);
amcmahon@med.usc.edu (APMM)

Present address: †Department of Cell Biology and Physiology, University of North Carolina at Chapel Hill, Chapel Hill, United States

Competing interests: The authors declare that no competing interests exist.

**Abstract** A normal endowment of nephrons in the mammalian kidney requires a balance of nephron progenitor self-renewal and differentiation throughout development. Here, we provide evidence for a novel action of ureteric branch tip-derived Wnt11 in progenitor cell organization and interactions within the nephrogenic niche, ultimately determining nephron endowment. In *Wnt11* mutants, nephron progenitors dispersed from their restricted niche, intermixing with interstitial progenitors. Nephron progenitor differentiation was accelerated, kidneys were significantly smaller, and the nephron progenitor pool was prematurely exhausted, halving the final nephron count. Interestingly, RNA-seq revealed no significant differences in gene expression. Live imaging of nephron progenitors showed that in the absence of *Wnt11* they lose stable attachments to the ureteric branch tips, continuously detaching and reattaching. Further, the polarized distribution of several markers within nephron progenitors is disrupted. Together these data highlight the importance of Wnt11 signaling in directing nephron progenitor behavior which determines a normal nephrogenic program.
DOI: https://doi.org/10.7554/eLife.40392.001

## Introduction

The developing mammalian kidney contains three distinct progenitor niches in the active nephrogenic zone. They consist of the mesenchymal nephron and interstitial progenitor cells and the epithelial ureteric progenitor cells (*McMahon, 2016*). The progenitor niches are self-renewing and give rise to the majority of cell types that make up the mature kidney (*Kobayashi et al., 2008*; *Self et al., 2006*; *Kobayashi et al., 2014*; *Costantini, 2012*). There is a distinct organization of these cell types within the nephrogenic niche. Nephron progenitors tightly cap ureteric branch tips throughout the arborization of the ureteric epithelial-derived collecting duct system, separating the ureteric branch

tips from interstitial progenitors which lie at the kidney cortex. The interstitial progenitors also infiltrate the space between adjacent nephron progenitor caps (*McMahon, 2016*). This distinct niche organization is maintained throughout the course of kidney development, suggesting it may be significant for proper nephrogenesis. A recent study uncovered the dynamic movements of nephron progenitors within and between niches when not tightly associated with the branch tip (*Combes et al., 2016*). Control of nephron progenitor behavior through yet unknown mechanisms is likely critical for niche maintenance.

Genetic studies and cell culture experiments have provided extensive evidence for a complex interplay of signaling interactions amongst these progenitor populations through Wnt, Bmp, and Fgf mediated pathways (*Majumdar et al., 2003*; *Nagy et al., 2016*; *Carroll et al., 2005*; *Park et al., 2007*; *Karner et al., 2011*; *Luo et al., 1995*; *Dudley et al., 1999*; *Blank et al., 2009*; *Tomita et al., 2013*; *Muthukrishnan et al., 2015*; *Brown et al., 2011*; *Barak et al., 2012*; *Motamedi et al., 2014*). Amongst Wnt-family members, Wnt9b and Wnt11 produced by ureteric epithelial cells, regulate distinct programs in overlying nephron progenitors. Wnt9b is required for both the expansion of uncommitted nephron progenitors and the commitment of a subset of nephron progenitors to enter a nephron forming program, in conjunction with the branching growth of the ureteric epithelial network (*Carroll et al., 2005*; *Park et al., 2007*; *Karner et al., 2011*). The co-requirement for the Wnt pathway transcriptional co-activator, β-catenin, in these events suggest Wnt9b acts through the canonical Wnt pathway (*Park et al., 2007*; *Karner et al., 2011*; *Wiese et al., 2018*), although Wnt9b also has roles in tubular morphogenesis via non-canonical planar cell polarity pathways (*Karner et al., 2009*). Unlike *Wnt9b* which shows lower expression in branch tips immediately adjacent to nephron progenitors than in tip-derived cells of non-branching stalks, *Wnt11* expression is highly restricted to branch tips, from the earliest stages of kidney development (*Majumdar et al., 2003*; *Kispert et al., 1996*; *Combes et al., 2017*). Expression of *Wnt11* is positively regulated by nephron progenitor and potentially interstitial progenitor-derived Gdnf, acting through the Ret receptor pathway in ureteric branch tips (*Majumdar et al., 2003*; *Costantini and Shakya, 2006*; *Magella et al., 2018*). Wnt11 signaling acts back on nephron progenitors to maintain a level of *Gdnf* expression sufficient for normal branching morphogenesis of the ureteric epithelium (*Majumdar et al., 2003*). Wnt11 generally works through non-canonical mechanisms in regulating developmental processes such as convergent extension and cardiogenesis (*Heisenberg et al., 2000*; *Tada and Smith, 2000*; *Nagy et al., 2010*; *Zhou et al., 2007*). Non-canonical Wnts control cellular behaviors including motility, adhesions, and rearrangements of the cytoskeleton independent of β-catenin mediated transcriptional regulation (*Wiese et al., 2018*; *van Amerongen, 2012*). Whether Wnt11 acts through similar non-canonical mechanisms in the developing kidney remains to be determined.

Recently, analysis of the *Wnt11* mutant phenotype on the C57BL/6 background allowed for the survival of a subset of mutants until adulthood (*Nagy et al., 2016*). In these animals, tubular morphology was disrupted and glomerular cysts observed, both likely culprits for the compromise in kidney function. The expression of *Wnt11* in the tubular epithelium of both postnatal mice and adults may be partially responsible for this phenotype. Alternatively, alterations to the expression of *Six2*, *Wnt9b*, *Gdnf*, and *Foxd1* were seen in developing *Wnt11*[-/-] kidneys suggesting the downregulation of these genes could contribute to the phenotype (*Nagy et al., 2016*). Despite further informative characterization of the *Wnt11* mutant phenotype, a fundamental understanding of actions immediately downstream of Wnt11 signaling during kidney development is still lacking.

Our examination *Wnt11* mutant kidneys revealed a novel requirement for Wnt11 signaling in the organization of nephron progenitors within the nephrogenic niche. Here, we present evidence that the tight organization of nephron progenitors around ureteric branch tips is characterized by a Wnt11-dependent interaction of nephron progenitors with underlying epithelial cells through stable cytoplasmic extensions. Following the loss of this dynamic interplay, the balance between maintenance and commitment of nephron progenitors is offset towards commitment, prematurely depleting the nephron progenitor reserve, resulting in smaller kidneys with fewer nephrons. Taken together with studies of *Wnt9b*, the new findings indicate nephron progenitors are likely to integrate both canonical (Wnt9b) and non-canonical (Wnt11) signaling pathways in developmental regulation of the nephrogenic niche.

## Results

### Wnt11 is required for nephron progenitor niche organization and proper nephron endowment

Previous reports suggested that branching morphogenesis is disrupted in *Wnt11* mutants, leading to smaller kidneys (*Majumdar et al., 2003*). However, an understanding of molecular mechanisms underpinning the phenotype were lacking. Additionally, the postnatal lethality precluded analyses of adult phenotypes. We took advantage of the *Wnt11* knockout-first reporter allele available from the EUCOMM/KOMP repository to analyze the mutant phenotype in greater detail (*Skarnes et al., 2011*). The insertion of a cassette with an alternate splice-acceptor terminates the transcript after exon two and an Internal Ribosomal Entry Site (IRES) promotes lacZ reporter expression. This allele (*Wnt11$^{tm1a}$*) disrupts the *Wnt11* transcript shortly after the N-terminal signal peptide required for ligand secretion (*Figure 1—figure supplement 1A,B*; *Willert and Nusse, 2012*). We compared β-galactosidase activity to *Wnt11* in situ expression patterns at embryonic day 11.5 (E11.5), E15.5, and postnatal day 2 (P2) and confirmed the reporter recapitulates *Wnt11* expression in the ureteric tips throughout kidney development (*Figure 1—figure supplement 1C,D*). The line was maintained on a C57BL/6J background which enabled survival of the *Wnt11* mutant (*Wnt11$^{tm1a/tm1a}$*) mice into adulthood.

We utilized antibodies against Six2, a definitive regulator of nephron progenitor self-renewal (*Kobayashi et al., 2008*; *Self et al., 2006*), together with a pan-cytokeratin antibody that recognizes the ureteric epithelium (Krt8/18) (*Oosterwijk et al., 1990*), to obtain an overview of niche organization in wholemount kidney preparations (*Figure 1A*). Strikingly, analysis at E15.5 showed a marked dispersion of Six2$^+$ nephron progenitors in nephrogenic niches of *Wnt11* mutants (*Figure 1B*). Consistent with wholemount observations, tissue section revealed Six2$^+$ nephron progenitors were no longer tightly clustered around ureteric branch tips in *Wnt11* mutants (*Figure 1C*, right panels). The disorganization of nephron progenitors was evident as early as E11.5, shortly after outgrowth of the ureteric bud, the initiating morphogenetic event in mammalian kidney development (*McMahon, 2016*). Six2$^+$ cells were less organized and more rounded in appearance compared to the tightly elongated, aligned nephron progenitors present in wild type kidneys (*Figure 1C*, left panels). By E13.5, *Wnt11$^{tm1a/tm1a}$* nephron progenitors showed a similar scattering to that observed at E15.5 (*Figure 1C*, middle panels). At birth, large subsets of progenitor niches had completely disappeared, three days prior to the normal loss of the Six2 +nephron progenitor population and cessation of nephrogenesis (*Figure 1D,E*; *Hartman et al., 2007*; *Rumballe et al., 2011*).

Quantitative analysis showed a statistically significant decrease in the kidney-to-body weight ratio (20%), an overall reduction in kidney volume (37%), and a 41% deficit reduction of branch tips in *Wnt11$^{tm1a/tm1a}$* kidneys (*Figure 1—figure supplement 2A–D*; *Figure 1—source data 1*). We predicted that the premature depletion of nephron progenitors would affect adult nephron endowment. Indeed, glomerular counts performed six weeks after birth showed only 50% of the normal nephron number in *Wnt11* mutant mice (*Figure 2A*; *Figure 2—source data 1*). Despite this significant deficit, *Wnt11$^{tm1a/tm1a}$* kidney volume was only reduced by 28% compared to wild type animals (*Figure 2D*; *Figure 2—source data 1*) reflecting compensatory hypertrophic growth, evident in the increased size of glomeruli (18%) and renal corpuscles (27%) in *Wnt11* mutants (*Figure 2B,C,E*; *Figure 2—source data 1*).

### Quantitative analyses reveal differences in niche metrics

Next, we performed quantitative analyses on wholemount immunostained kidneys at E15.5 to identify differences in niche metrics. Samples were subjected to optical projection tomography (OPT) and confocal imaging for subsequent quantitative analyses (*Figure 3A*; *Short et al., 2014*). The nephrogenic niches within *Wnt11$^{tm1a/tm1a}$* mutant kidneys contained a larger number of Six2$^+$ nephron progenitors compared to wild type kidneys: a 1.3-fold increase from an average of 365 to 466 Six2 +cells per niche (*Figure 3B*; *Figure 3—source data 1*). However, despite the increase in cell number, there was no statistically significant change in nephron progenitor niche (cap) volume (*Figure 3C*; *Figure 3—source data 1*). Similar to nephron progenitors, the number of cells per ureteric tip was increased 1.5-fold in *Wnt11$^{tm1a/tm1a}$* kidneys compared to controls (Wt = 162, Het = 169, Mut = 248; *Figure 3D*; *Figure 3—source data 1*). Accordingly, tip volumes were

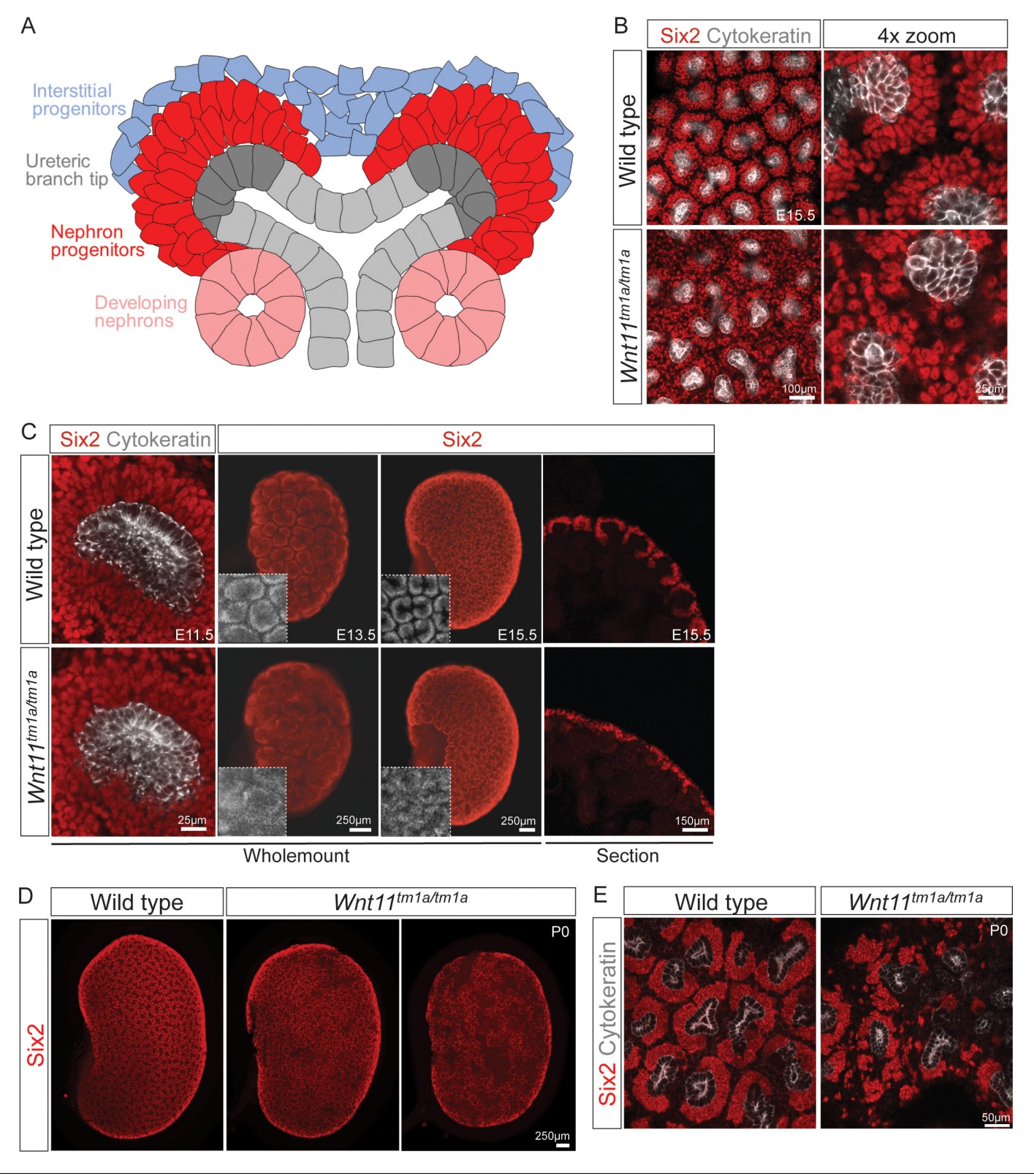

**Figure 1.** *Wnt11* mutants have persistent loose, disorganized nephron progenitor niches that prematurely dropout. (**A**) Schematic of the nephrogenic niche. Wnt11 is secreted by the ureteric branch tip cells. (**B**) Wholemount immunostained kidneys were analyzed at E15.5. Confocal images show that *Wnt11tm1a/tm1a* Six2 +nephron progenitors (red) are dispersed from the ureteric branch tips (grey) and rounded in appearance compared to wild type cells. A 4x zoom in the far-right panel highlights the differences in cellular distribution and morphology. (**C**) Wholemount immunostained wild type and
*Figure 1 continued on next page*

*Figure 1 continued*

*Wnt11* mutant kidneys were analyzed at E11.5, E13.5, and E15.5. *Wnt11*$^{tm1a/tm1a}$ Six2 +nephron progenitors are more rounded and less organized than wild type counterparts beginning at E11.5 (view shows slice from a z-stack). The disorganized phenotype is clear in whole kidney views (maximum intensity projection) at E13.5 and E15.5. Insets highlight the disorganization in whole kidney views. The far-right panel shows a cryosection of E15.5 wild type and *Wnt11* mutant kidneys immunostained for Six2, highlighting the dispersed phenotype is also evident in section. (D) Wholemount immunostains of P0 kidneys show the persistent disorganized phenotype and associated premature dropout of Six2 +nephron progenitor niches (red). (E) High resolution confocal views highlight the premature dropout of Six2 +nephron progenitor niches (red) around cytokeratin +ureteric epithelium (grey) in *Wnt11*$^{tm1a/tm1a}$ kidneys.

DOI: https://doi.org/10.7554/eLife.40392.002

The following source data and figure supplements are available for figure 1:

**Source data 1.** Quantitation of P0 kidney metrics.
DOI: https://doi.org/10.7554/eLife.40392.005

**Figure supplement 1.** The *Wnt11*$^{tm1a}$ allele recapitulates expression of *Wnt11*.
DOI: https://doi.org/10.7554/eLife.40392.003

**Figure supplement 2.** Kidney metrics at P0 reveal deficits in *Wnt11* mutants.
DOI: https://doi.org/10.7554/eLife.40392.004

increased 1.6-fold in mutants (*Figure 3E*; *Figure 3—source data 1*). OPT analysis, which provides a clear overview of ureteric tree metrics, indicates *Wnt11* mutant kidneys were smaller than control kidneys. *Wnt11*$^{tm1a/tm1a}$ kidney ureteric tree volume was reduced to 67% of wild type (*Figure 3F*; *Figure 3—source data 2*). Accordingly, ureteric tree length was 30% shorter in *Wnt11* mutants (*Figure 3—figure supplement 1C*). In line with the reduced kidney size, mutant kidneys had 33% fewer ureteric tips (Wt = 609, Het = 532, Mut = 407; *Figure 3G*; *Figure 3—source data 2*). However, overall branching patterns were similar among genotypes suggesting normal but slightly delayed morphogenesis (*Figure 3—figure supplement 1A*). These data are consistent with *Wnt11* mutants having smaller, developmentally younger kidneys (*Short et al., 2014*). Several of the kidneys analyzed (5/8) had prematurely terminated branch tips internally (*Figure 3—figure supplement 1A,B*) which may indicate that these tips had lost their association with the nephron progenitors or inductive signals at some point during development.

Interestingly, the *Wnt11* mutant kidneys had a greater number of differentiating structures per tip. On average, there were >2 developing nephrons per *Wnt11*$^{tm1a/tm1a}$ tip while controls contained <2 (*Figure 3H*; *Figure 3—source data 1*). The increased number of developing nephrons were biased towards renal vesicles and Stage 4 + nephrons versus comma- or s-shaped bodies (*Figure 3I*). Taken together, the *Wnt11* mutant phenotype suggests a decrease in branching growth consistent with earlier reports (*Majumdar et al., 2003*), and an accelerated differentiation of nephron progenitors relative to the stage of ureteric epithelial outgrowth.

Given the role for Wnt11 in regulating the Gdnf-Ret signaling axis in the first few branching events (*Majumdar et al., 2003*), we examined whether tip identity was maintained in *Wnt11* mutant kidneys. Etv4 and *Wnt11* itself (marked by β-gal in the tm1a allele), maintained their expected tip restricted expression (*Figure 4A*, data not shown). Additionally, we analyzed other markers which restrict nephrogenic niche identity. The matrix protein fibronectin (Fn1) is largely excluded from the nephron progenitor niche of wild type kidneys, though high levels were present within adjacent interstitial progenitors. This is consistent with RNA-seq data which indicates *Fn1* is expressed by interstitial and not nephron progenitors (*O'Brien et al., 2016*). In *Wnt11* mutants, fibronectin also accumulated in the nephron progenitor zone suggesting a possible movement of interstitial cells into this region (*Figure 4B*). Indeed, Foxd1$^+$ interstitial progenitors interdigitated with Six2 +nephron progenitors (*Figure 4C*). Approximately four times as many Foxd1$^+$ cells contacted ureteric branch tips in *Wnt11*$^{tm1a/tm1a}$ kidneys (*Figure 4D*; *Figure 4—source data 1*). Six2$^+$ cells were also more frequently observed in interstitial progenitor space (*Figure 1B,C*; *Figure 4C*). Whereas the nearest neighbor of any Six2$^+$ nephron progenitor cell in wild type kidneys was most likely another Six2$^+$ cell, Six2$^+$ cells were as likely to have a Foxd1$^+$ neighboring cell in E15.5 *Wnt11* mutant kidneys (*Figure 4E*; *Figure 4—source data 1*). These findings indicate that Wnt11 is required for stratification of progenitor cell types within the nephrogenic niche.

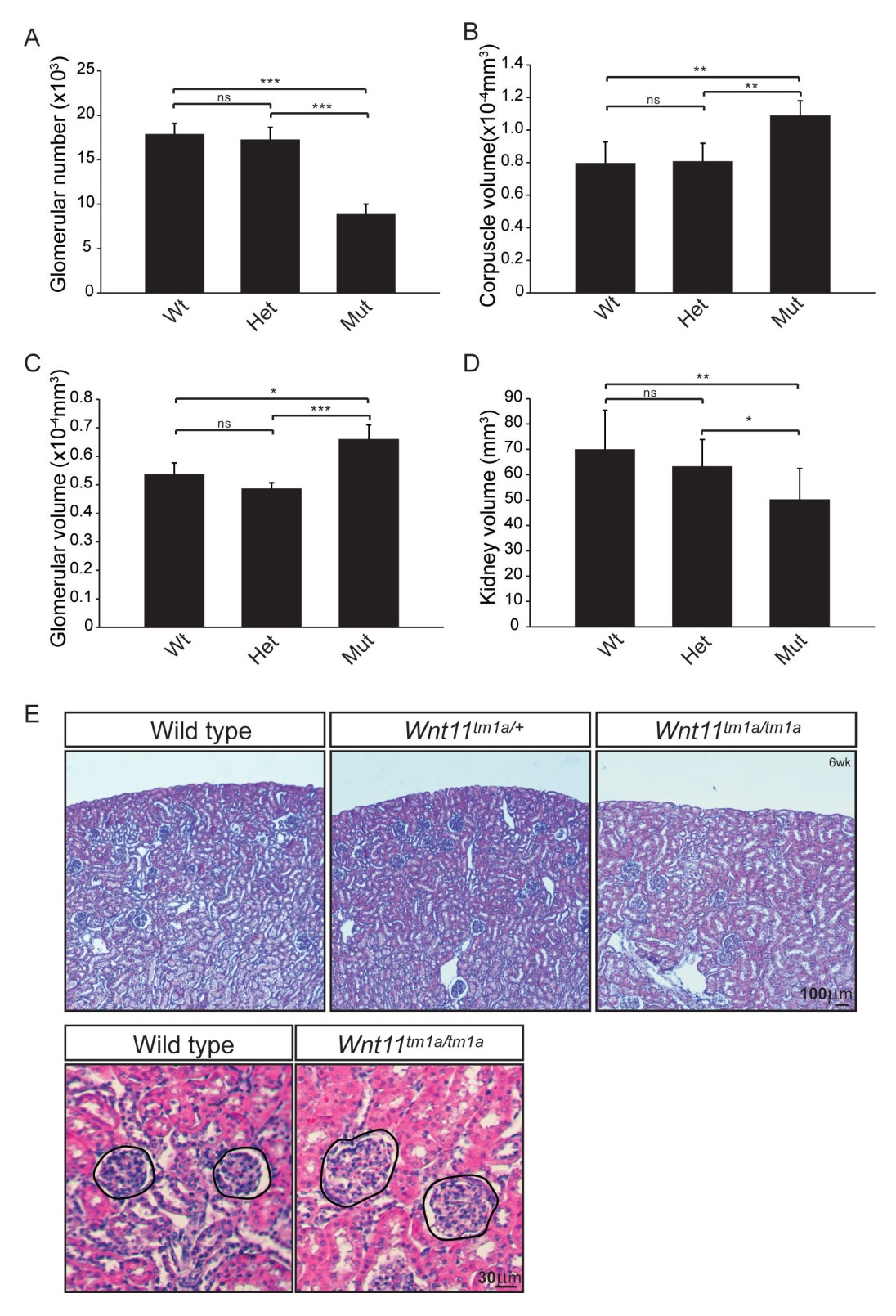

**Figure 2.** Nephron numbers are reduced in *Wnt11tm1a/tm1a* adults leading to compensatory hypertrophy. (**A**) The physical disector/fractionator method was used to estimate glomerular number in 6 week old animals. A significant reduction is found in *Wnt11* mutants. (**B**) Renal corpuscle volume is increased in *Wnt11* mutant kidneys. (**C**) Glomerular volume is larger in *Wnt11tm1a/tm1a* kidneys. (**D**) Estimated kidney volume is reduced in *Wnt11* mutants. (**E**) Histological sections stained with hematoxylin and eosin highlight that overall morphology is preserved, although glomerular/corpuscle

*Figure 2 continued on next page*

*Figure 2 continued*

volume is increased in the *Wnt11* mutants. All error bars represent SEM. All significance values were determined by t-test. ns = p > 0.05, * = p < 0.05, ** = p < 0.01, *** = p < 0.001. n = 6 for each genotype.

DOI: https://doi.org/10.7554/eLife.40392.006

The following source data is available for figure 2:

**Source data 1.** Quantitation of adult kidney metrics.

DOI: https://doi.org/10.7554/eLife.40392.007

## Gene expression analyses highlight a non-canonical role of Wnt11 in regulating cell behavior

To determine whether loss of *Wnt11* altered gene expression within nephron progenitors, we introduced a *Six2* promoter/enhancer driven GFP transgene (*Six2TGC$^{tg}$*; *Kobayashi et al., 2008*) into the *Wnt11* mutant background. Utilizing the *GFP* allele, we performed fluorescence-activated cell sorting (FACS) in order to transcriptionally profile the purified nephron progenitors by RNA-sequencing. At E15.5, no significant changes were observed in the expression of key nephron progenitor factors such as *Six2*, *Fgf20*, and *Gdnf* in *Wnt11* mutant nephron progenitors (*Figure 4F*, *Supplementary file 1*). None of the genes with a greater than 1.5-fold change and reasonable expression level (RPKM >10) showed consistent results across replicates (*Supplementary file 1*). The up-regulation of *Egr1-4*, *Fos*, *Fosb*, *Jun*, and *Junb* in one *Wnt11* mutant nephron progenitor sample most likely reflects a variable, isolation-associated stress response (*Supplementary file 1*; *Adam et al., 2017*). To avoid isolation artifacts, we compared whole kidney RNA-seq profiles at E15.5. Again, we observed only modest transcriptional changes between wild type and *Wnt11* mutant kidneys (<1.5 fold, *Figure 4F*; *Supplementary file 2*). The marked reduction in expression of the targeted *Wnt11* allele provides an internal control validating the approach and confirming detection is sensitive enough to reveal gene expression changes within a small subset of cells in the kidney (*Figure 4F*, *Supplementary file 2*). Thus, Wnt11 likely regulates cell behaviors rather than nephron progenitor gene regulatory programs through β-catenin-dependent transcriptional regulation, consistent with a non-canonical Wnt-signaling action.

## Phenotypic analyses of putative receptors for Wnt11 reveal functional redundancy

To gain insight into the molecular pathway underlying Wnt11 action, we characterized the expression of known Wnt receptors and co-receptors in nephron progenitors. RNA-seq analysis of E16.5 isolated nephron progenitors (*O'Brien et al., 2018*) identified >20 possible candidates with an RPKM >1 and about half with an RPKM >10 (*Figure 4—figure supplement 1A*). Top candidates included *Fzd2*, *Fzd7*, *Ryk*, *Ptk7*, and *Ror2*. *Ror2* is highly expressed in nephron progenitors at E13.5 though down-regulated by E15.5 and engages Wnt5a in the regulation of early kidney development (*Nishita et al., 2014*; *Yun et al., 2014*). *Fzd2* and *Fzd7* exhibit redundant functions in convergent extension and ventricular septum closure; the latter is partially mediated by Wnt11 (*Yu et al., 2012*). We confirmed the expression of these two *Fzd* receptors within the nephron progenitors by in situ hybridization and examining β-galactosidase activity from the mutant alleles (*Figure 4—figure supplement 1B–D*; *Yu et al., 2012*; *Yu et al., 2010*). Ryk is a Wnt co-receptor with roles in the developing nervous system (*Fradkin et al., 2010*). Ptk7 is a Wnt receptor with established roles in cell migration and polarization, including morphogenesis of the Wolffian duct (*Lu et al., 2004*; *Xu et al., 2018*; *Xu et al., 2016*).

To address the potential for Wnt11 action through these receptors, we examined the organization of Six2 +progenitor niches in homozygous mutant kidneys at E15.5 for several receptor mutants. While neither *Fzd2* or *Fzd7* mutant kidneys showed a phenotype (*Figure 4—figure supplement 1E*), a weakly penetrant phenotype (1 in 3 embryos) - a mild disruption of nephron progenitor organization (*Figure 4—figure supplement 1G*) - was observed in *Fzd2$^{-/-}$; Fzd7$^{+/-}$* kidneys. Unfortunately, the lethality of *Fzd2$^{-/-}$; Fzd7$^{-/-}$* mutants prior to active kidney development precluded complete removal of both these receptors (*Yu et al., 2012*). *Ptk7* mutant kidneys displayed a weak Six2$^+$ nephron progenitor disorganization in one of two embryos examined (*Figure 4—figure supplement 1E, F*) while progenitor organization was normal in *Ryk* and *Ror2* mutants (*Figure 4—figure supplement*

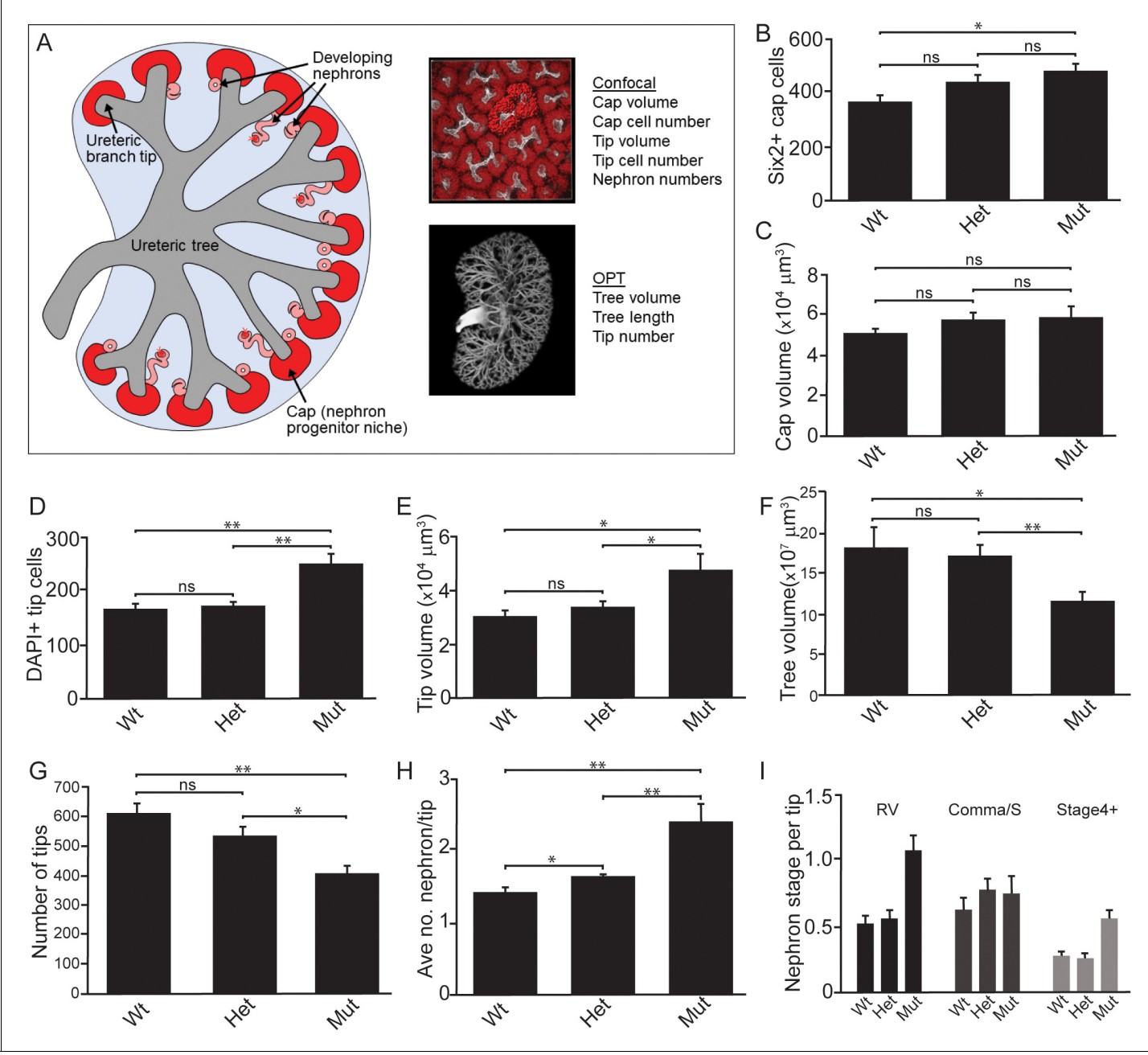

**Figure 3.** Quantitative analyses reveal significant alterations to niche metrics and accelerated nephrogenesis in *Wnt11*[tm1a/tm1a] kidneys. (**A**) Schematic of the developing kidney highlighting the niche and structure metrics quantified after imaging by either confocal or optical projection tomography (OPT). E15.5 wholemount immunostains were performed with α-Six2 and α-pan cytokeratin antibodies and kidneys subsequently imaged by confocal microscopy for analyses in B-E, H, and I or OPT for analyses in F-G. (**B**) The number of Six2 +nephron progenitors per niche were quantified and reveal an increase in *Wnt11* mutants. (**C**) The overall volume of each nephron progenitor niche (cap) is not significantly different between wild type and *Wnt11* mutants. (**D**) The number of DAPI +cells were quantified in each cytokeratin +ureteric branch tip niche and were increased in *Wnt11* mutants. (**E**) Ureteric branch tip volumes were measured and are larger in *Wnt11* mutants, correlating with the increase in tip number. (**F**) Ureteric tree volume was quantified and is reduced in *Wnt11*[tm1a/tm1a] kidneys. (**G**) Quantitation of ureteric branch tip number reveals a significant decrease in *Wnt11*[tm1a/tm1a] kidneys. (**H**) The average number of nephrons per ureteric branch tip are increased in *Wnt11* mutant kidneys. (**J**) Classification of the developing nephron structures associated with each tip highlight a bias in *Wnt11* mutants towards renal vesicles (RV) and stage 4 + nephrons versus comma/s-shaped bodies. All error bars represent SEM. All significance values were determined by t-test. ns = p > 0.05, * = p < 0.05, ** = p < 0.01, *** = p < 0.001. n = 6 kidneys of each genotype for confocal analyses. N = 8 of each genotype for OPT analyses.

DOI: https://doi.org/10.7554/eLife.40392.008

The following source data and figure supplement are available for figure 3:

*Figure 3 continued*

**Source data 1.** E15.5 confocal analyses.
DOI: https://doi.org/10.7554/eLife.40392.010
**Source data 2.** E15.5 OPT analyses.
DOI: https://doi.org/10.7554/eLife.40392.011
**Figure supplement 1.** Branching patterns are similar among genotypes despite smaller ureteric trees and premature branch truncations in *Wnt11* mutants.
DOI: https://doi.org/10.7554/eLife.40392.009

*1E*). In light of these results and the substantial number of Wnt receptors putatively expressed by the nephron progenitors (*Figure 4—figure supplement 1A*), functional compensation likely hinders the identification of a definitive receptor for Wnt11.

## Stable nephron progenitor attachments and intrinsic polarization are disrupted in Wnt11 mutants

To better understand how cellular behaviors in the *Wnt11* nephrogenic niche would result in their dispersed phenotype, we combined a tamoxifen inducible Six2-driven Cre strain (*Six2CE*; (*Kobayashi et al., 2008*)) and the *R26^{mTmG}* reporter mouse (*Muzumdar et al., 2007*) to label the cell membrane of nephron progenitors in wild type and *Wnt11* mutant backgrounds. Tamoxifen injection 24 hr prior to tissue collection resulted in labeling of a small subset of nephron progenitors, allowing precise tracking of individual cellular dynamics. In fixed tissue sections, nephron progenitors were packed tightly around the tips and elongated perpendicular to the surface of branch tips with an average width-to-length ratio of 0.450 (*Figure 5A,B*; *Figure 4—source data 1*). Wild type GFP$^+$ nephron progenitors sent out prominent membranous projections to the underlying cytokeratin positive ureteric branch tips, often extending over the surface of the epithelium (*Figure 5A*). In contrast, *Wnt11* mutant nephron progenitors were more rounded with an increased width-to-length ratio of 0.565 (*Figure 5A,B*; *Figure 4—source data 1*). Further, progenitors also appeared to make fewer and less extensive contacts with the ureteric bud in *Wnt11* mutant kidneys (*Figure 5A,C*). Quantifying the frequency and surface area of GFP$^+$ nephron progenitor contacts with the ureteric branch tips demonstrated that >80% of nephron progenitors exhibit an attachment versus 45% in *Wn11* mutants (*Figure 5E*; *Figure 5—source data 1*), and the area of contact was reduced by 30% in mutant kidneys (*Figure 5D,F*). Overall there was no significant difference in the length of these extensions or their angle of protrusion relative to the ureteric bud between either attached or unattached cells of either genotype (*Figure 5—figure supplement 1A–D*; *Figure 4—source data 1* ).

To obtain a dynamic insight into nephron progenitor behavior, we performed live imaging in kidney organ culture (*Figure 6A*; *Figure 6—video 1*). GFP$^+$ nephron progenitors were divided into three categories based on initial analysis of cell motility, cell contacts, and cell extensions over 14.5 hr of imaging: (1) attached cells with a wide, stable attachment and those in the process of foot retraction or extension (2) detached cells, and (3) cells undergoing mitosis and dispersal (*Figure 6B*). For live imaging, we utilized Wnt11 heterozygous kidneys as controls. We found no significant difference in the behavior of wild type and heterozygous nephron progenitors, and this allowed to utilize a mating strategy that optimized the number of kidneys we could image and reduced overall animal numbers required. Heterozygous control GFP$^+$ nephron progenitors showed consistent, stable attachments to the ureteric bud detaching as cells underwent cell division (*Figure 6A,B,D*; *Figure 6—video 1*). In contrast, *Wnt11^{tm1a/tm1a}* cells displayed frequent rounds of attachment, detachment, and reattachment (*Figure 6A,B,D*; *Figure 6—video 1*). Whether initially attached or unattached, nephron progenitors displayed $\geq$4 fold the number of attachments or reattachments in *Wnt11* mutant versus control kidneys (*Figure 6C*; *Figure 6—source data 1*). Nephron progenitors in *Wnt11* mutants also displayed a larger displacement over time and an increased velocity, but no significant difference in meandering (*Figure 6—figure supplement 1A–C*). In summary, the absence of stable epithelial tip attachment by nephron progenitors in *Wnt11* mutant kidneys correlates with a dispersal of the nephron progenitor niche, accelerated nephrogenesis, and the premature loss of the nephrogenic niche.

Elongation of control nephron progenitors towards ureteric branch tips suggests cells exhibit a polarized behavior in response to Wnt11 producing cells. Therefore, we analyzed several indicators

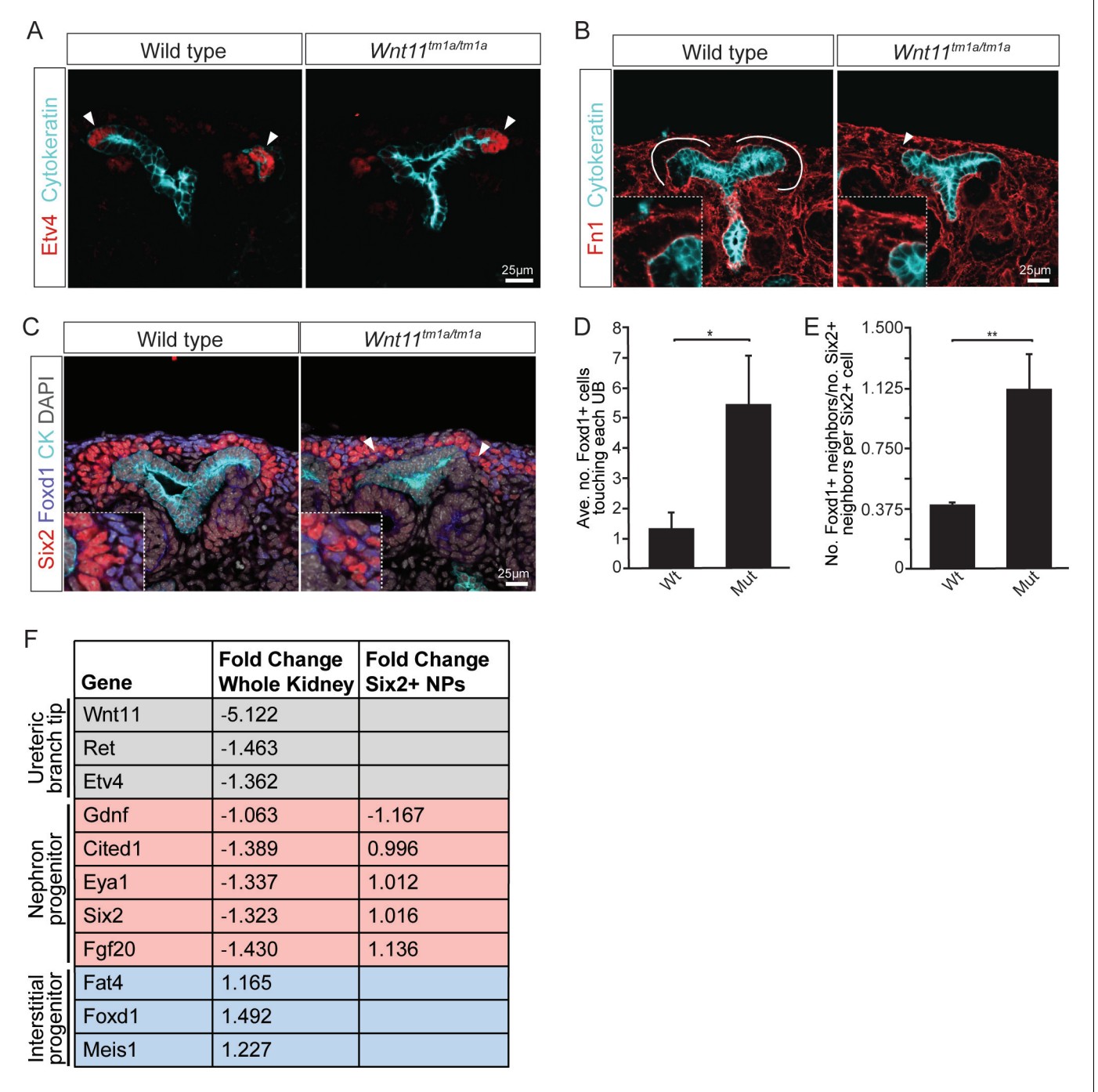

**Figure 4.** Nephron progenitors intermix with interstitial progenitors but no significant changes in gene expression are observed. (**A**) E15.5 kidneys were sectioned and immunostained for the tip marker Etv4 (red) and cytokeratin (cyan). Distinct ureteric branch tip domains are still present in *Wnt11* mutants as indicated (arrowheads). (**B**) E15.5 sections were stained for the matrix protein fibronectin (Fn1; red) and cytokeratin (cyan). Note the exclusion of fibronectin from the nephron progenitor niche in wild type animals (line marks boundary between the nephron progenitors and interstitial progenitors). In *Wnt11^tm1a/tm1a* kidneys the fibronectin boundary is disrupted and staining observed in the nephron progenitor niche (arrowheads). Insets show zoomed view of the progenitor niche. (**C**) Immunolocalization of Foxd1 +interstitial progenitors (blue) in conjunction with Six2 +nephron progenitors (red) at E15.5 reveals mixing of the two cell populations in *Wnt11* mutant kidneys (insets show zoomed view of cell mixing). Foxd1 +cells can infiltrate (arrowheads) the nephron progenitor niche and are found near the ureteric branch tips (cyan). (**D**) Quantitation of tissue sections immunostained for both Six2, Foxd1, and cytokeratin. There is an increase in the number of Foxd1 +cells touching the ureteric branch tips in Wnt11 mutants. 30 ureteric tip domains from n = 3 biological replicates were quantified. (**E**) Quantitation of Six2 cell neighbors. The number of Foxd1 +cells touching a Six2 +cell is divided by the number of Six2 +cells touching the same cell. In *Wnt11* mutants a Six2 +cell is just as likely to have as many

*Figure 4 continued on next page*

*Figure 4 continued*

Foxd1 +neighbors as Six2 +neighbors. Three biological replicates were quantified, 10 Six2 +cells per sample. (**F**) Fold-changes associated with RNA-seq of either whole kidneys or Six2 +cells from wild type and *Wnt11* mutant kidneys. The fold-change was calculated from the average of n = 6 for each genotype in whole kidney analysis and n = 3 for each genotype in the nephron progenitor analysis. Example genes which define each progenitor population (ureteric branch tip, nephron progenitor, and interstitial progenitor) are shown. No significant changes (>1.5 fold change) are observed. All error bars represent SEM. All significance values were determined by t-test. ns = p > 0.05, * = p < 0.05, ** = p < 0.01, *** = p < 0.001.

DOI: https://doi.org/10.7554/eLife.40392.012

The following source data and figure supplement are available for figure 4:

**Source data 1.** Quantitation of nephron progenitor metrics.
DOI: https://doi.org/10.7554/eLife.40392.014
**Figure supplement 1.** Several Wnt receptors are expressed in the nephron progenitors and display weakly penetrant phenotypes upon deletion.
DOI: https://doi.org/10.7554/eLife.40392.013

of nephron progenitor polarity. During directed cell migration, Golgi are oriented to the leading edge of the cell. Analysis of wild type E15.5 kidneys showed an asymmetry in the cellular location of the Golgi apparatus to the distal half of nephron progenitors, furthest from the ureteric epithelium (*Figure 7A*). However, a bias in Golgi positioning within nephron progenitors was not observed in *Wnt11* mutant kidneys (*Figure 7A–C*). As a second measure, we analyzed the localization of integrin α8. Integrin α8 (Itga8) is synthesized by nephron progenitors and binds nephronectin (Npnt), a matrix protein secreted by the ureteric epithelium, a critical interaction for mammalian kidney development (*Müller et al., 1997*; *Linton et al., 2007*). Integrin α8 is polarized on the surface of nephron progenitors concentrating in the membrane closest to the ureteric branch tip (*Figure 7D,E*; (*Uchiyama et al., 2010*)). However, this polarization was lost and integrin α8 was dispersed throughout nephron progenitors in *Wnt11* mutant kidneys (*Figure 7E*). Lastly, we examined desmin (Des), an intermediate filament protein. In contrast to nephron progenitors in control kidneys which showed an extensive accumulation of proximally localized desmin within 5 μm of ureteric branch tips, a substantial desmin accumulation was observed outside of this 'near-tip zone' in *Wnt11* mutant kidneys (*Figure 7F,G*). These data highlight several aspects of cell polarity within mesenchymal nephron progenitors that are dependent on Wnt11 production by ureteric branch tips.

## Discussion

Our detailed analysis of the kidney phenotype in *Wnt11* mutants reveals a critical role for Wnt11 signaling in the organization and cellular dynamics of nephron progenitors in the nephrogenic niche of the mammalian kidney. In the absence of Wnt11 signals from the ureteric branch tips, the nephron progenitors cannot maintain stable attachments to the tips. As a result, nephron progenitors mix with the adjacent interstitial progenitors, undergo an accelerated differentiation, and are prematurely depleted, leading to a significant reduction in nephron endowment. The nephrogenic niche is continually moving which presents a problem for keeping together coherent, spatially-defined communities of interacting progenitor types. Our data indicates that Wnt11-dependent interactions are critical for maintaining progenitor niche integrity over the course of mammalian kidney development. Human nephrogenesis persists for a much longer period than that of the mouse: 30 weeks versus 11 days (*Little and McMahon, 2012*). Maintaining an organized nephron progenitor niche throughout that period, to generate an average of one million nephrons per kidney (*Hoy et al., 2003*), presents a challenge on a different scale. Interestingly, SIX2+ nephron progenitors display a tight association with the ureteric branch tips in the human kidney (*O'Brien et al., 2016*; *Lindström et al., 2018*) suggesting a conservation amongst mammalian species in mechanisms organizing the nephrogenic niche. Consistent with a role for WNT11 in these events, *WNT11* shows a similar ureteric branch tip-restricted expression in the developing human kidney (*Rutledge et al., 2017*).

### Wnt11 loss leads to disorganized nephron progenitors, smaller kidneys, and reduced nephron numbers

We previously showed that Wnt11 regulates the early ureteric branching program with *Wnt11* mutants generating a smaller kidney, consistent with our current analyses (*Majumdar et al., 2003*). The phenotype was attributed to the disruption of a reciprocal signaling axis between Wnt11, Gdnf,

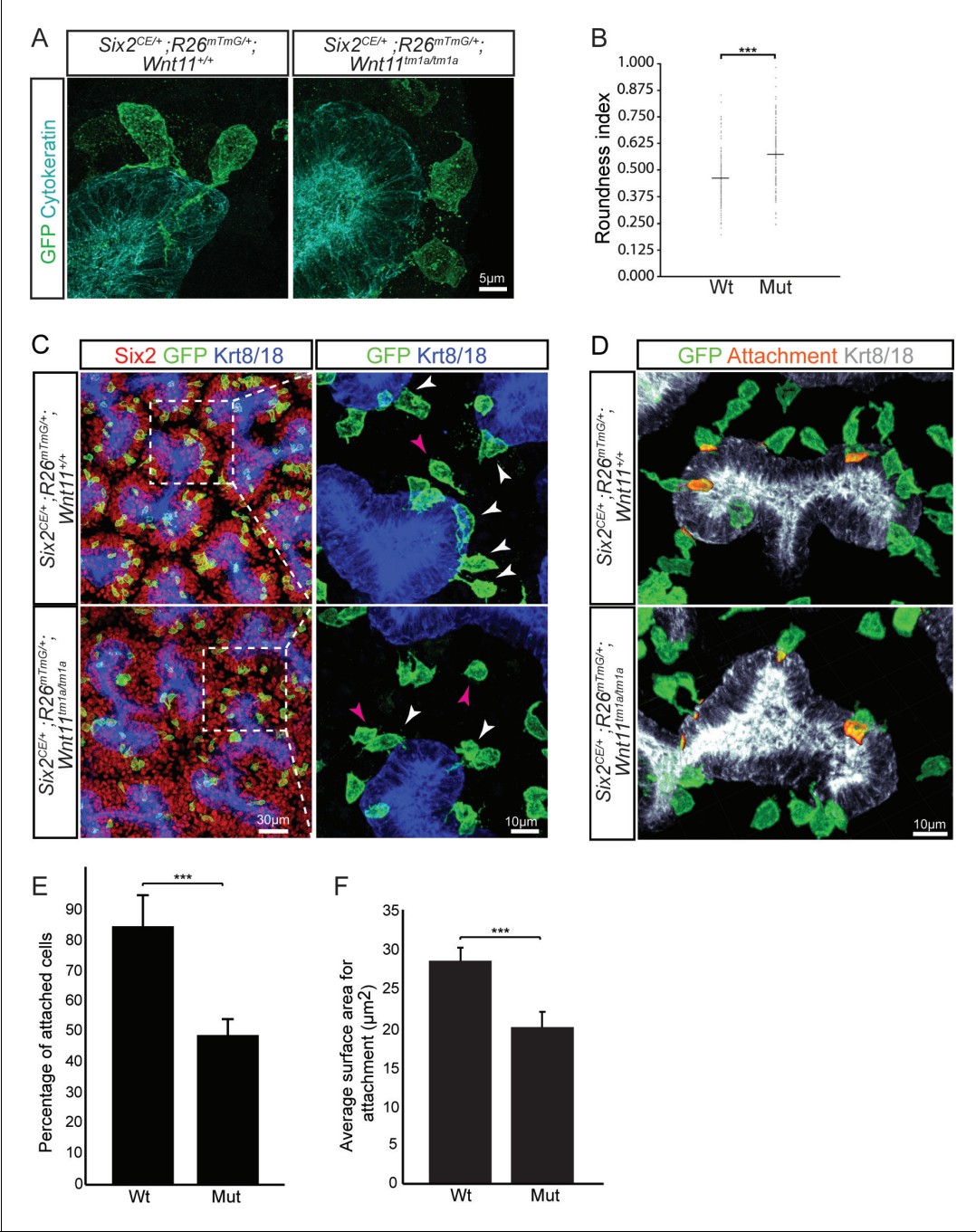

**Figure 5.** *Wnt11* mutants show a significant reduction in membranous attachments of nephron progenitors to the ureteric branch tip. (**A**) Sections from E15.5 *Six2$^{CE/+}$; R26$^{mTmG/+}$; Wnt11$^{+/+}$* and *Six2$^{CE/+}$; R26$^{mTmG/+}$; Wnt11$^{tm1a/tm1a}$* kidneys (recombination induced 24 hr prior) were immunostained for GFP (green) and cytokeratin (cyan) to visualize the GFP +cells in relation to the ureteric branch tips. Nephron progenitors make long, membranous projections which attach to the ureteric bud. Less extensive projections make contact with the ureteric branch tip in *Wnt11* mutants and appear rounder. (**B**) Quantitation of GFP +cell roundness from samples in A) (length/width = roundness index) reveals that *Wnt11* mutant nephron progenitors are rounder. n = 100 cells for each genotype. (**C**) Wholemount immunostains were carried out on E15.5 *Six2$^{CE/+}$; R26$^{mTmG/+}$; Wnt11$^{+/+}$* and *Six2$^{CE/+}$; R26$^{mTmG/+}$; Wnt11$^{tm1a/tm1a}$* kidneys (recombination induced 24 hr prior) for Six2 (red), GFP (green), and cytokeratin 8/18 (Krt8/18; blue). Confocal images show the dispersion of GFP +cells within the Six2 +nephron progenitor niche. Magnified views point to attached (white arrowhead) and detached (pink arrowhead) cells, with more detached cells present in *Wnt11* mutants. (**D**) Kidneys similar to those from C) showing the overlay of GFP (green) and Krt8/18 (grey) signal as an area of attachment (orange). More extensive areas of attachment are observed in wild type kidneys. (**E**) Quantitation of attachments from samples similar to C) showing a significant reduction in the percentage of attached cells in Wnt11 mutants. n = 200–400 cells from each of 4 biological replicates were analyzed. (**F**) Quantitation of attachment area as shown in D). The average surface area of attachments is reduced in

*Figure 5 continued on next page*

*Figure 5 continued*

*Wnt11* mutants. All error bars represent SEM. All significance values were determined by t-test. ns = p > 0.05, * = p < 0.05, ** = p < 0.01, *** = p < 0.001.

DOI: https://doi.org/10.7554/eLife.40392.015

The following source data and figure supplement are available for figure 5:

**Source data 1.** Quantitation of nephron progenitor attachments.

DOI: https://doi.org/10.7554/eLife.40392.018

**Figure supplement 1.** Quantitation of cellular extensions reveals no significant differences between wild type and *Wnt11* mutant nephron progenitors.

DOI: https://doi.org/10.7554/eLife.40392.016

and Ret. This was based on the observations that (1) at E12.5 *Wnt11* mutants showed reduced *Gdnf* expression in the overlying capping mesenchyme populations as assayed by in situ hybridization, (2) ectopic Gdnf elevated *Wnt11* expression in the ureteric epithelium, and (3) a genetic interaction was observed between *Wnt11* and *Ret* mutant alleles (*Majumdar et al., 2003*). Together these data supported a model wherein Gdnf signaling by mesenchymal progenitors acting through the Ret receptor within ureteric branch tips promoted *Wnt11* expression, and Wnt11 signaling to the underlying nephron progenitors promoted Gdnf production and normal epithelial branching. However, by E13.5, no difference was observed in *Gdnf* expression between wild type and *Wnt11* mutant kidneys suggesting a transient requirement for this positive feedback loop (*Majumdar et al., 2003*). In line with these results, our RNA-seq data shows no significant change in expression of *Gdnf* (or any other gene) in E15.5 nephron progenitors from *Wnt11* mutant kidneys. Similarly, *Npnt* mutants show reduced *Gdnf* expression at E11.5 but normal levels at E13.5 (*Linton et al., 2007*). In contrast, the novel phenotype we describe here, a dispersed, disorganized nephron progenitor population, is persistent throughout development, and it is most likely this and not the early reduction in branching that is the major contributor to the premature loss of nephrogenic niches and the marked reduction in nephron number in the adult kidney. Consistent with this view, half of the nephrons are generated from nephron progenitor cells between P0 and P4 (*Short et al., 2014*), the time when nephrogenic niches are substantially reduced in *Wnt11* mutants.

## Nephron progenitors display extensive membranous processes and dynamic behavior

Through our limited labeling of nephron progenitors, we observe a direct dynamic interplay between nephron progenitors and the ureteric tips that harbor the progenitors for the ureteric epithelial network of the collecting system. These results are in line with prior insights into the cellular dynamics within nephrogenic niche (*Combes et al., 2016*). In this previous study, wild type nephron progenitor attachments to the ureteric tips were assayed, but fewer stable attachments were observed than in the current study. These prior observations utilized a cytoplasmic fluorescent protein to label cells, whereas the current analysis used a membrane localized reporter that facilitates better resolution of membrane processes. The elaboration of membranous protrusions is a common behavior of mesenchymal cells and has been noted previously for nephron progenitors. Studies dating back to 1975 have identified by light and electron microscopy that such cellular protrusions are sent out by nephron progenitors towards the ureteric epithelium (*Lehtonen, 1975*; *Minuth and Denk, 2016*). Our studies suggest these cell-cell contacts may facilitate the tracking of nephron progenitor cells with outgrowing branch tips and forming a polarized cell barrier of nephron progenitors that separates nephron and interstitial progenitor niches. In other developing systems, notably the Drosophila imaginal disc and vertebrate limb, long membrane extensions have been linked to transmission and reception of signals within distinct signaling pathways (*Sanders et al., 2013*; *Kornberg and Roy, 2014*). Whether the observed extensions play a role in active signaling mechanisms within the nephrogenic niche warrants further exploration.

The ureteric epithelium is the source of Wnt9b which has been shown to play a critical role in inducing nephron progenitor cells (*Carroll et al., 2005*). Interestingly, despite the dispersal of nephron progenitors away from ureteric branch tips, we observe an accelerated differentiation of nephron progenitors that would appear to be at odds with what is normally a Wnt9b-mediated differentiation process. However, it is difficult to predict the outcome on the basis of our current

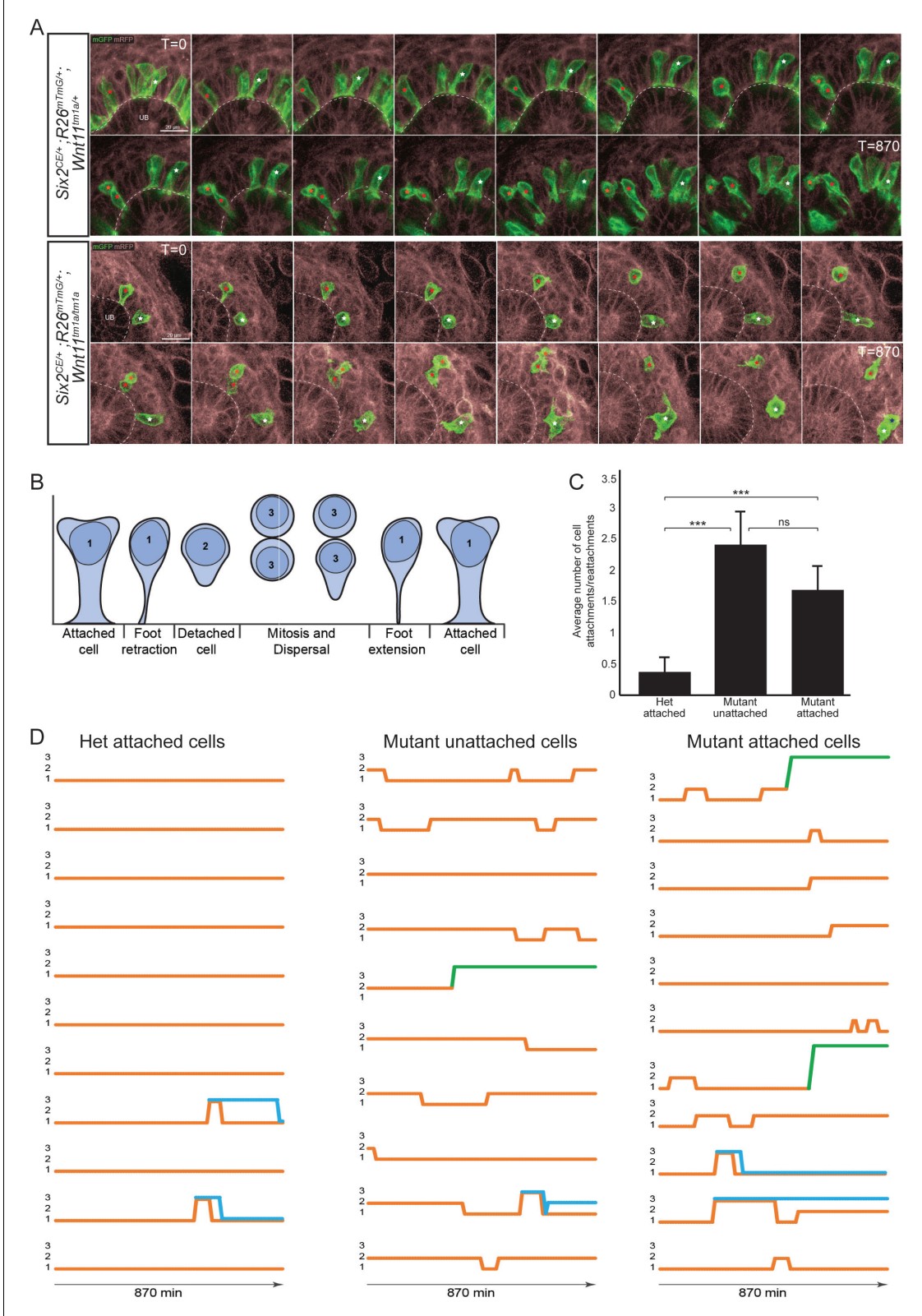

**Figure 6.** The nephron progenitors of *Wnt11* mutants display dynamic attachments and reattachments to the ureteric bud. (A) Still images from *Six2^{CE/+}*; *R26^{mTmG/+}*; *Wnt11^{tm1a/+}* and *Six2^{CE/+}*; *R26^{mTmG/+}*; *Wnt11^{tm1a/tm1a}* kidney explant cultures live-imaged over 870 min (14.5 hr). GFP +cells are in green, the tdTomato +membrane of all other cells is in grey. Two cells are marked with an asterisk (red or white) in each genotype and followed across the time. The cells marked with red asterisk go through a cell division, the white cells do not. In the control kidneys, the white cell stays attached

*Figure 6 continued on next page*

*Figure 6 continued*

throughout. The red cell detaches briefly to divide and quickly reattaches. In *Wnt11* mutants, both cells display dynamic attachments and reattachments. (B) Schematic of the classification scheme utilized to define cellular dynamics of the nephron progenitors. (C) Quantitation of the average number of attachments and reattachments of cells during live imaging. All het control cells generally begin attached and display few attachments/reattachments. *Wnt11* mutant cells, whether they were attached or detached when imaging began, both display numerous attachments/reattachments. (D) Representative tracks of 10–11 individual cells classified as in B) over the course of live imaging. 11 control and 33 mutant cells were analyzed in total. Orange tracks highlights the transition between stages. Control cells stayed attached and only detached to divide. Mutant cells, whether initially attached or detached, show dynamic movements. Blue track = new cell from a division. Green = cell migrated out of the imaging field. All error bars represent SEM. All significance values were determined by t-test. ns = p > 0.05, * = p < 0.05, ** = p < 0.01, *** = p < 0.001.
DOI: https://doi.org/10.7554/eLife.40392.019

The following video, source data, source code and figure supplement are available for figure 6:

**Source data 1.** Analyses of nephron progenitor movements.
DOI: https://doi.org/10.7554/eLife.40392.021
**Source code 1.** Matlab script for analyses of movements.
DOI: https://doi.org/10.7554/eLife.40392.022
**Figure supplement 1.** *Wnt11* mutant nephron progenitors show greater displacement and velocities, but meander similarly to wild type cells.
DOI: https://doi.org/10.7554/eLife.40392.020
**Figure 6—video 1.** Time lapse of *Wnt11* mutant and control kidneys reveal differences in nephron progenitor dynamics.
DOI: https://doi.org/10.7554/eLife.40392.023

understanding of the complex roles for signaling and cell organization. For example, Wnt9b is also linked to progenitor expansion (*Karner et al., 2011*), together with localized production of a large number of other signals from nephron progenitors, interstitial progenitors and ureteric progenitors (*McMahon, 2016*) and it is the integration of all of these that will determine cell behavior. The physical disruption of normal cell boundaries is likely to significantly modify these signaling inputs. Wnt11 could also mediate Wnt9b effects on nephron progenitors, normally working to prevent their induction. Differences in cell shape and polarization may dictate their differentiation capacity. Alternatively, a reduction in Wnt11 signals could allow differentiation of nephron progenitors through the action of an additional commitment regulator away from the tip.

## Wnt11 acts as a non-canonical regulator of nephron progenitor cellular behavior

Wnt11 signals predominantly through non-canonical pathways; for example, in *Xenopus* and zebrafish development, where Wnt11 controls cellular behaviors necessary for convergent extension during embryogenesis (*Heisenberg et al., 2000*; *Tada and Smith, 2000*; *Marlow et al., 2002*; *Ulrich et al., 2005*), or in mammalian heart development, where Wnt11 directs cell behaviors responsible for outflow tract morphogenesis (*Zhou et al., 2007*). Cells respond to non-canonical Wnt signals typically by rearranging the cell polarity, cytoskeletal organization, adhesive properties, and cell motility. Signal transduction activates downstream signaling cascades which modify cell activities without substantial alteration in transcriptional programs, although JNK- and NFAT-mediated transcription can be activated in some cases (*van Amerongen, 2012*). Our findings support a non-canonical action for Wnt11 in the kidney, though evidence is predominantly the absence of a transcriptional readout on removing Wnt11 signaling and the associated cell phenotypes displayed by nephron progenitor cells in *Wnt11* mutant kidneys: a loss of polarity in the organization of cells and cellular contents, and reduced stabilization of membrane contacts suggestive of reduced adhesive contacts that likely underscores the enhanced cell mobility of nephron progenitors. A future analysis of the requirement of key non-canonical pathway components will provide additional resolution to pathway action.

Given the observed nephron progenitor phenotypes and the fact that these cells directly contact Wnt11 expressing cells in ureteric branch tips, nephron progenitors are the likely direct target of Wnt11 action. However, proof will again require an insight into Wnt11 transducing mechanisms. We performed extensive analysis of several Wnt receptor mutants to obtain an insight into the transducing receptor. Though the results are not definitive, single or combinatorial mutations in *Ptk7*, *Fzd2*, and *Fzd7* resulted in weak, and weakly penetrant, kidney phenotypes resembling the *Wnt11* mutant linking these receptors to Wnt11 signal transducing processes. Redundancy amongst

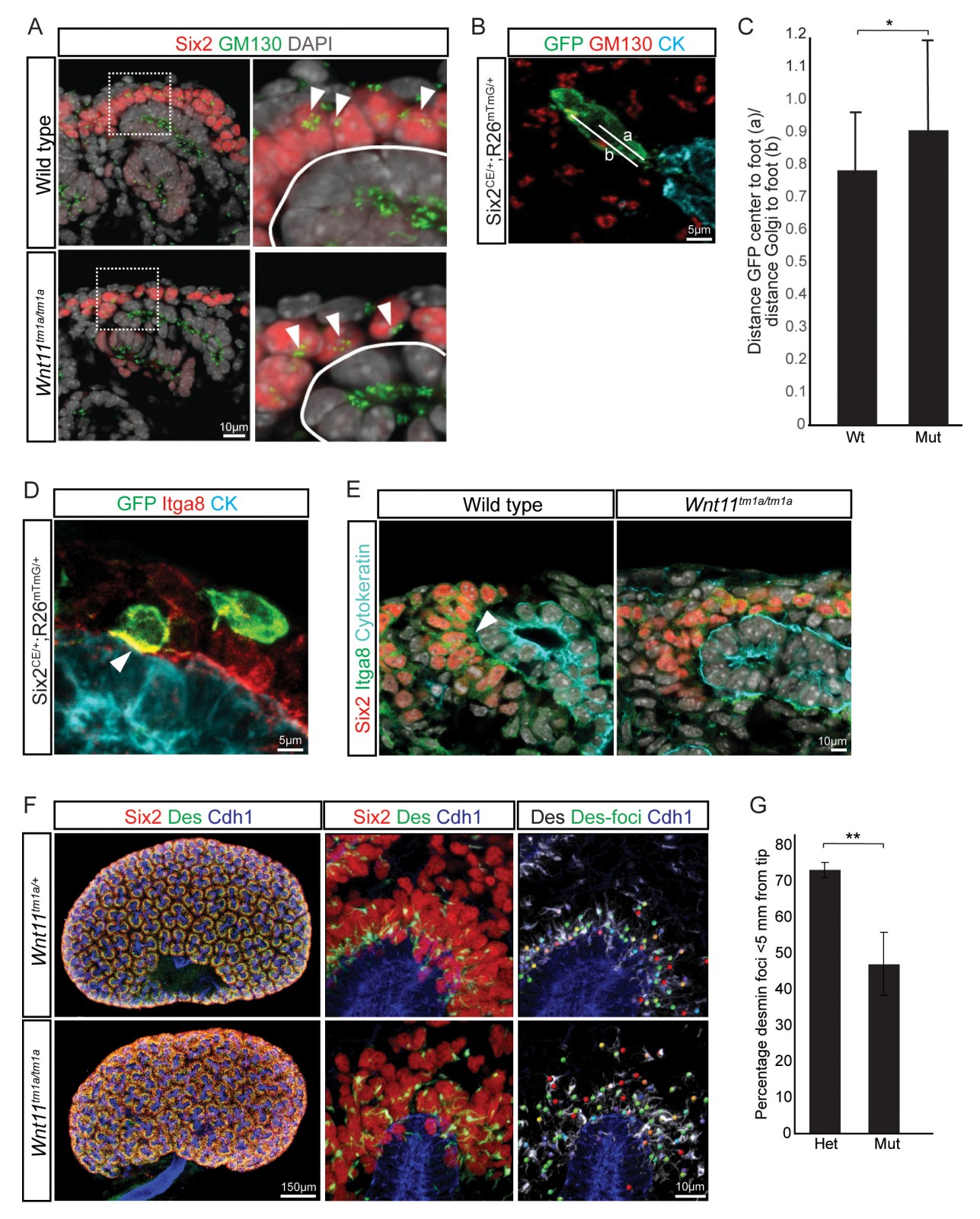

**Figure 7.** Nephron progenitor polarity is disrupted in *Wnt11* mutants. (**A**) E15.5 kidney sections were immunostained for Six2 (red), GM130 (Golgi; green) and DAPI (grey). The Golgi (white arrowheads) show a polarization to the distal end of nephron progenitors in wild type kidneys and this polarization is disrupted in *Wnt11* mutants where they are localized closer to the ureteric branch tip (white outline). (**B**) Staining for GFP (green), GM130 (red), and cytokeratin (cyan) highlighting the normal polarization of Golgi within the nephron progenitor at E15.5. The letter 'a' marks the distance from

*Figure 7 continued on next page*

Figure 7 continued

the cell center to the foot of the cell (contact with tip) and 'b' marks the distance from the Golgi to the foot. (C) Ratio of the distances described in B) for wild type and mutant cells. Wild type cells show a smaller ratio indicating the Golgi lie farther from the ureteric tip than Wnt11 mutant cells, supporting their polarized nature and the loss in Wnt11 mutants. Three biological replicates and ~20 cells from each replicate were quantified. (D) E15.5 kidney sections immunostained for GFP (green), integrin a8 (Itga8; red) and cytokeratin (CK, cyan). Image shows the overlap of GFP +signal with Itga8 in cells close to the tip (white arrowhead), suggesting polarization of Itga8. (E) E15.5 kidney sections immunostained for Six2 (red), Itga8 (green), and cytokeratin (cyan). Arrowhead points to Itga8 polarization towards the ureteric branch tip in wild type kidneys, which is lost in Wnt11 mutants. (F) E15.5 wholemount immunostains for Six2 (red), desmin (Des; green), and cadherin 1 (Cdh1; blue) show the polarization of desmin foci (Des-foci) toward the ureteric branch tip which is disrupted in Wnt11 mutant kidneys. Since the desmin stain appears aster-like with wispy projections, the central focal point (foci) were identified for ease of distance quantitation. Foci were automatically located by Imaris imaging software as the most intense focal point of the desmin stain. (G) Quantitation of the percentage of desmin foci located greater than 5 μm from the ureteric tip in each genotype showing the greater dispersion from the tip in Wnt11 mutants. 385 foci from five control tips and 353 foci from four tips were analyzed. All error bars represent SEM. All significance values were determined by t-test. ns = p > 0.05, * = p < 0.05, ** = p < 0.01, *** = p < 0.001.

DOI: https://doi.org/10.7554/eLife.40392.024

vertebrate Wnt receptors is more the rule than the exception. For example, *Fzd2* and *Fzd7* are known to be functionally redundant in convergent extension and cardiac development (*Yu et al., 2012*), *Fzd1* and *Fzd2* are partially redundant in processes such as palate closure (*Yu et al., 2010*), while *Fzd4* and *Fzd8* are both implicated in modulating branching morphogenesis in the developing kidney (*Ye et al., 2011*). Clarity on the cellular target and receptor and downstream mechanisms of Wnt11 signaling will entail the analysis of multiple alleles with the additional requisite that their activity can be specifically removed within cell types of interest in the nephrogenic niche, most notably the nephron progenitors.

In conclusion, these studies have uncovered novel aspects of nephron progenitor behavior in response to Wnt signaling. The findings underscore the importance of progenitor niche organization to proper kidney development and nephron endowment. Further insights into Wnt11 action will help inform efforts to recreate kidneys ex vivo through the generation of nephrogenic niches that have replicative, self-organizing capability.

## Materials and methods

### Mouse strains

All mouse handling and husbandry were performed according to guidelines issued by the Institutional Animal Care and Use Committees (IACUC) at the University of Southern California and after approval (protocol #11893). *Six2TGC^{tg}* and *Six2CE* mice were generated previously as described (RRID:IMSR_JAX:009606, RRID:IMSR_JAX:032488; (*Kobayashi et al., 2008*)). *Fzd2* (*B6;129-Fzd2^{tm1.1Nat}/J*, Stock No: 012821; RRID:IMSR_JAX:012821), *Fzd7* (*B6;129-Fzd7^{tm1.1Nat}/J*, Stock No: 012825; RRID:IMSR_JAX:012825), *Ror2^c* (*B6;129S4-Ror2^{tm1.1Meg}/J*, Stock No: 018354; RRID:IMSR_JAX:018354), and *Rosa26^{mTmG}* (*B6.129(Cg)-Gt(ROSA)26Sor^{tm4(ACTB-tdTomato,-EGFP)Luo}/J*, Stock No: 007676; RRID:IMSR_JAX:007676) were all purchased from Jackson Labs (JAX). *Wnt11^{tm1a(KOMP)Wtsi}* Knockout First ES cells (JM8.N4 line; Targeting project CSD47978; RRID:IMSR_KOMP:CSD47978-1a-Wtsi) were purchased from the EUCOMM/KOMP Repository. ES cells were injected into albino-B6 blastocysts by the Genome Modification Facility (Harvard University) to generate founders. All mouse strains were maintained on the C57BL/6J background (JAX Stock No: 000664; RRID:IMSR_JAX:000664). *Ryk* mice were previously described (RRID:MGI:2667559; (*Halford et al., 2000*)). *Ptk7* mice were previously described (RRID:MGI:3047812; (*Lu et al., 2004*)). Genotyping for all JAX strains was performed as described for each line on the JAX website (www.jax.org). *Wnt11^{tm1a}* mice were genotyped with the following primers: Wild type allele: F-ACCTGCTTGACCTGGAGAGA, R-AAGTGTTATTCGGGCCACTG; Mutant allele: F-ACCTGCTTGACCTGGAGAGA, R-CCAACTGACCTTGGGCAAGAACAT.

### In situ hybridizations and β-galactosidase stains

In situ hybridizations were performed as described on GUDMAP for sections and wholemounts. (www.gudmap.org; McMahon group protocols). Probes for *Wnt11*, *Fzd2* and *Fzd7* are those previously utilized (www.gudmap.org; *Wnt11* assay ID: N-H6DT; *Fzd2* assay ID: N-H5QG; *Fzd7* assay ID:

N-EDX6). For β-galactosidase assays, whole embryos or kidneys were collected, fixed for two hours in 4% paraformaldehyde, and kidneys were either kept whole or cryosectioned; whole embryos were cryosectioned. Standard protocols were utilized for X-gal staining of tissues.

## Immunostains

For wholemounts, kidneys at the appropriate stage (E11.5, E13.5, E15.5, or P0) were collected and fixed for 20 min in 4% paraformaldehyde. They were washed 3x in PBS (5 min each) and put into block solution for 1–2 hr (block = PBS + 10% sheep serum +0.1% Triton X-100). Block was removed and primary antibodies diluted in block solution were added to each. Kidneys were incubated overnight at 4°C. The next day samples were washed for 5 min at room temperature with PBS + 0.1% TX100. They were subsequently washed 3x for ~2 hr each wash at 4°C. Samples were incubated with the appropriate species-specific AlexaFluor secondary antibody (Life Technologies) diluted 1:250 in block solution overnight at 4°C. The samples were washed again as previously described. Samples were counterstained with DAPI if necessary. Wholemount kidneys were imaged with a ZEISS Axio Zoom.V16 Fluorescence Stereo Zoom microscope or Leica SP8 inverted confocal. For kidney sections, samples were collected, fixed for 1–2 hr in 4% paraformaldehyde, immunostained, and imaged with a Nikon Eclipse 90i epi-fluorescent microscope, Zeiss LSM 780 inverted confocal microscope, or Leica SP8 inverted confocal as previously described (*Park et al., 2012*). For *Figure 7A*, Huygens deconvolution software was utilized post-imaging. Antibodies utilized in this study include: Six2 (Proteintech, 11562–1-AP, RRID:AB_2189084, 1:1000 for section, 1:250 for wholemount), pan-cytokeratin (Sigma, C2931, RRID:AB_258824, 1:250), GFP (Aves Labs, GFP-1020, RRID:AB_10000240, 1:500), integrin α8 (R and D Systems, AF4076, RRID:AB_2296280, 1:500), Etv4 (Abgent, AP6642B, RRID:AB_1967633, 1:500), Krt8/18 (Abcam, ab53280, RRID:AB_869901, 1:500), Cdh1 (BD Transduction Laboratories, 610182, RRID:AB_397581, 1:500), Foxd1 (Santa Cruz Biotechnology, sc-47585, RRID:AB_2105295, 1:1000), fibronectin (Sigma, F3648, RRID:AB_476976, 1:500), desmin (DAKO, M0760, RRID:AB_2335684, 1:500), GM130 (BD Transduction Laboratories, G65120/610822, RRID:AB_398141, 1:500), and β-galactosidase (Biogenesis, 4600–1409, RRID:AB_2314513, 1:1000).

## Quantitation of kidney metrics

Wholemount immunostained kidneys were imaged by either confocal or optical projection tomography (OPT) as previously described (*Short et al., 2014*). Quantitative analyses from confocal images (Six2 +nephron progenitors, DAPI +tip numbers, cap volume, tip volume, nephron number per tip, nephron type) and OPT images (tree volume, tree length, number of tips; additionally analyses from Tree surveyor can be found in *Figure 3—source data 2*) were performed as previously described (*Short et al., 2014*).

## Nephron counts and associated adult phenotypic analysis

Kidneys were collected from 6 week old male and female animals. Six kidneys were analyzed from each genotype (wild type, *Wnt11$^{tm1a/+}$*, and *Wnt11$^{tm1a/tm1a}$*). Nephron counts were performed using the physical disector/fractionator combination (*Cullen-McEwen et al., 2012a*). Estimations of glomerular volume and renal corpuscle volume were obtained by dividing the total volume of the kidney occupied by either glomerular tuft or renal corpuscles by the number of glomeruli counted (*Cullen-McEwen et al., 2012a*; *Cullen-McEwen et al., 2012b*). Kidney volume was estimated by the Cavalieri principle. Paraffin sections (5 μm) were stained by hematoxylin and eosin and imaged by brightfield microscopy on a Nikon Eclipse 90i.

## Live confocal imaging of nephron progenitor cells in Wnt11 heterozygous and mutant kidneys

Pregnant female mice were injected with tamoxifen (1 mg tamoxifen/40 gm of animal) 24 hr prior to embryo collection. E11.5 kidneys were dissected and embedded into 40 μl of DMEM:Matrigel (Corning, BD354277) at a 1:1 ratio. The kidneys in Matrigel were then placed on a MatTek 35 mm culture dish (MatTek P35G-0–20 C) and the Matrigel was allowed to set for 45 min at 37°C, 5% $CO^2$. Subsequently, 2 ml of kidney culture media (DMEM +10% fetal bovine serum +1X Glutamax +1% Penicillin/Streptomycin) was added to the dish. The kidneys were cultured overnight at 37°C, 5% $CO^2$. During the overnight culture the kidneys flattened against the MatTek glass bottom surface. The

dish was placed within an Ibidi culture chamber (IBIDI) which was heated to 37°C and is gassed to 5% $CO^2$. The cultures were allowed to equilibrate for 1 hr prior to imaging. Single-cell tracking and imaging was performed using a Leica SP8 using a 40x objective (40x/1.30 Oil HC PL APO CS2). 512 × 512 images were captured at every 10 min through whole tips at step-interval of 3 μm in the z-direction. Laser lines and detectors were adjusted to for GFP, RFP, and transmitted light capture. Data-sets were captured over 16 hr periods with a 10 min interval. In total,>5 $Six2^{CE/+}$; $R26^{mTmG/+}$; $Wnt11^{tm1a/+}$ and > 4 $Six2^{CE/+}$; $R26^{mTmG/+}$; $Wnt11^{tm1a/tm1a}$ were imaged as described above. To reduce the number of kidneys and animals necessary as some kidneys would fail to grow or imaging would prove unsuccessful due to drift out of imaging plane/area, $Wnt11^{tm1a/+}$ animals were crossed to $Wnt11^{tm1a/tm1a}$ animals to produce only heterozygous or mutant animals for analyses. We found no significant difference between wild type and heterozygous nephron progenitor cell behavior. 87 het controls and 62 mutant cells were manually tracked in 3D using the Imaris Track (Bitplane, RRID: SCR_007370) function. Tracks were initially produced in Imaris and manually annotated to ensure accurate progression. We opted for manual tracking as the automated tracking function provided poor accuracy despite excellent signal-to-noise ratios in the samples. Because the tracked cells were located at the tip of a growing structure, the overall movement of the cells was heavily influenced by the general movement of the growing organ. To account for this, we therefore used drift correction to adjust for the frame-by-frame kidney growth.

Cell tracking was performed using the GFP reporter which localizes to cell-membranes. As the membrane display a higher level of movement compared to the cell's actual displacement, we triangulated the tracks to down sample membrane movements (*Figure 8*). These calculations were performed in MATLAB (MathWorks, RRID:SCR_001622; *Figure 6—source code 1*). This approach gave a visually improved accuracy for the cell-tracking. Cell movement was tracked from frame 0 but analyses were performed only from frame 10 and onwards (100 min) to ensure that no artifacts were introduced due to setting up the time-lapse capture. Analyses were then performed from frame 10 to frame 97 (870 min/14.5 hr).

## Stability of cell attachment/detachment from the ureteric bud tip

Using the time-lapse data captured for cell-tracking, cells were labelled as *attached* or *unattached* at frame 10. Attached meant that they had a clear membrane protrusion interacting with the ureteric branch tip. Unattached meant they did not have a protrusion in contact with the ureteric branch tip. Cells were then followed frame by frame and their status recorded as follows: 1 = attached, 2 = unattached, 3 = cell division, 4 = cell death, 5 = out of frame. 11 control cells and 33 mutant cells were monitored. Cells that died or moved out of frame were not included. The lower number for the control cells reflects the observation that no unattached cells were detected in any of the data-sets we analyzed at frame 10. In the mutant kidneys, approximately 50% of cells were unattached and therefore monitored. To statistically evaluate differences between wild type and mutant cells, each cell was scored for the number of times it changed status within categories 1–3. Student's t-tests were used to compare cell-types.

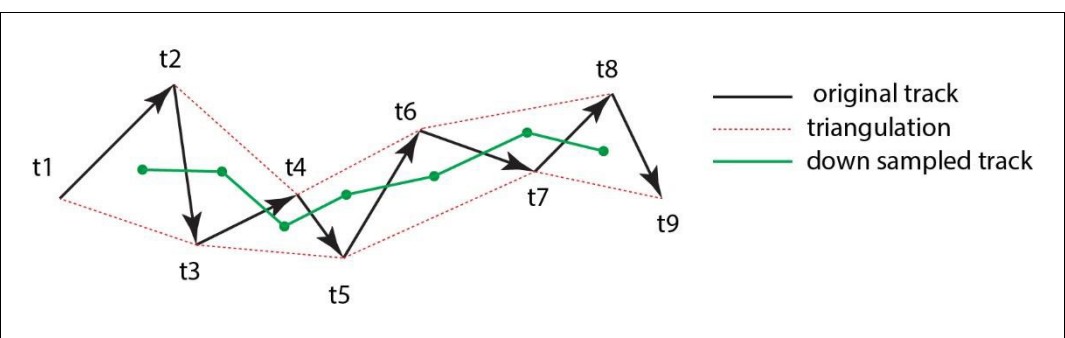

**Figure 8.** Triangulation to improve accuary of cell tracking. Schematic showing the method of triangulaton used to improve the accuracy of tracking nephron progenitors.

DOI: https://doi.org/10.7554/eLife.40392.025

## Three-dimensional reconstructions and analyses of cell attachment

Pregnant female mice were injected with 2 mg of tamoxifen per 40 g of mouse weight at E14.5 and $Six2^{CE/+}$; $R26^{mTmG/+}$; $Wnt11^{+/+}$ and $Six2^{CE/+}$; $R26^{mTmG/+}$; $Wnt11^{tm1a/tm1a}$ kidneys collected for analysis at E15.5. The kidneys were processed for wholemount stains for Six2, GFP, Krt8/18, to visualize nephron progenitor cells, membrane-GFP, and the ureteric branch tip, respectively. To determine the attachment size of wild type and mutant cells the following steps were taken utilizing Imaris software (Bitplane): 1. A surface was generated for the ureteric bud (Krt8/18 stain). To improve the signal edge homogeneity for the Krt8/18 stain, a 1 µm Gaussian blur was implemented on the ureteric bud stain. A surface was then constructed for the Krt8/18 stain with a 1 µm surface-detail resolution. 2. A 0.3 µm surface was built for the GFP channel. 3. The GFP surface was then used to mask the Six2 channel and the masked Six2 channel was blurred (Gaussian 0.5 µm) and the spot function used to count $Six2^+$/$GFP^+$ cells. 4. To identify GFP-membrane/Ureteric bud surface contacts the surface-surface contact tool was utilized. To remove noise, the generated surfaces were filtered at a threshold 0.5 µm$^2$. Remaining surfaces were analyzed for area of attachment (µm$^2$) in Imaris.

## Fluorescence activated cell sorting (FACS)

The $Six2TGC^{tg}$ allele was bred onto the $Wnt11^{tm1a}$ genetic background. E15.5 kidneys were dissected out and screened for GFP +fluorescence. Each GFP +kidney pair were put into a 1.5 ml tube and processed individually. Kidneys were dissociated with collagenase/dispase (Roche, 10269638001) at 37°C for approximately 10–15 min, resuspended in PBS containing 2% FBS and 10 mM EDTA, and filtered through 40 µm cell strainer (BD Falcon). GFP +cells were isolated by sorting with a BD FACSAria II. Cells were spun down and resuspended in buffer RLT from the Qiagen RNeasy micro kit. Genotyping was performed and samples from $Wnt11^{tm1a/tm1a}$; $Six2TGC^{tg/+}$ and $Wnt11^{+/+}$; $Six2TGC^{tg/+}$ animals were utilized for RNA-seq and differential gene expression analysis.

## RNA-seq

RNA was isolated from GFP +cells of the appropriate genotype using the QIAGEN RNeasy Micro Kit. Whole kidney RNA was isolated from either wild type or $Wnt11^{tm1a/tm1a}$ E15.5 kidneys with the QIAGEN RNeasy Mini Kit. Cells were resuspended and vortexed in buffer RLT and whole kidneys homogenized in buffer RLT using a motorized pestle. Whole kidneys were additionally centrifuged through a QIAshredder to shear genomic DNA and reduce viscosity. RNA was isolated following the manufacturer's instructions. Libraries were constructed from 20 to 50 ng of total RNA by the University of Southern California's Epigenome Center with the Illumina TruSeq RNA Library Prep Kit V2 following the manufacturer's instructions with following modifications: End-IT repair from Epicentre was used instead of TruSeq end repair (more favorable volumes). Half of adapter-ligation was amplified for a variable number of cycles depending on input amounts. KAPA Biosystems PCR Master Mix was utilized in place of Illumina's. Samples were run on the Illumina HiSeq 2000. Sequence files were aligned using TopHat v2.0.8b (RRID:SCR_013035) and Bowtie 2.1.0.0 (RRID:SCR_016368) by the USC Epigenome Center using default parameters. BAM files were processed to obtain RPKM values using the Partek Genomics Suite 6.6 software (RRID:SCR_011860, St. Louis, MO, USA). The Partek ANOVA in Partek Genomics Suite 6.6 software was utilized to find the differential expression of genes ('Differential expression analysis' function) between the two genotypes. Associated sequencing files can be found at Gene Expression Omnibus (https://www.ncbi.nlm.nih.gov/geo/, RRID:SCR_007303) under accession number GSE118334.

## Acknowledgements

We would like to thank Dr. Wange Lu (University of Southern California) and Dr. Xiaowei Lu (University of Virginia) for the *Ryk* and *Ptk7* embryonic kidney tissue that was used in our analyses, respectively. We also thank Dr. Seth Ruffins (University of Southern California) for help with microscopy.

## Additional information

### Funding

| Funder | Grant reference number | Author |
| --- | --- | --- |
| National Institute of Diabetes and Digestive and Kidney Diseases | DK054364 | Andrew P McMahon |
| Human Frontier Science Program | RGP0039/2011 | Ian Macleod Smyth<br>Melissa H Little<br>Andrew P McMahon |
| National Health and Medical Research Council | APP1002748 | Ian Macleod Smyth<br>Melissa H Little |
| Australian Research Council | DE150100652 | Alexander Nicholas Combes |
| National Institute of Diabetes and Digestive and Kidney Diseases | DK085959 | Lori L O'Brien |
| National Health and Medical Research Council | APP1063696 | Melissa H Little |
| National Health and Medical Research Council | APP1042093 | Melissa H Little |
| Australian Research Council | DP160103100 | Ian Macleod Smyth |
| The Eli and Edythe Broad Foundation | Broad Postdoctoral Fellowship | Lori L O'Brien |

The funders had no role in study design, data collection and interpretation, or the decision to submit the work for publication.

### Author contributions

Lori L O'Brien, Conceptualization, Data curation, Formal analysis, Supervision, Funding acquisition, Validation, Investigation, Visualization, Methodology, Writing—original draft, Writing—review and editing; Alexander N Combes, Conceptualization, Data curation, Formal analysis, Supervision, Funding acquisition, Validation, Investigation, Visualization, Methodology, Writing—review and editing; Kieran M Short, Conceptualization, Data curation, Software, Formal analysis, Validation, Investigation, Visualization, Methodology, Writing—review and editing; Nils O Lindström, Conceptualization, Data curation, Formal analysis, Supervision, Validation, Investigation, Visualization, Methodology, Writing—review and editing; Peter H Whitney, Formal analysis, Validation, Investigation, Visualization, Methodology, Writing—review and editing; Luise A Cullen-McEwen, Formal analysis, Validation, Investigation, Methodology, Writing—review and editing; Adler Ju, Formal analysis, Investigation, Writing—review and editing; Ahmed Abdelhalim, Software, Formal analysis, Investigation, Methodology, Writing—review and editing; Odyssé Michos, Formal analysis, Investigation, Methodology, Writing—review and editing; John F Bertram, Formal analysis, Supervision, Methodology, Writing—review and editing; Ian M Smyth, Conceptualization, Resources, Software, Formal analysis, Supervision, Funding acquisition, Methodology, Project administration, Writing—review and editing; Melissa H Little, Conceptualization, Resources, Formal analysis, Supervision, Funding acquisition, Methodology, Project administration, Writing—review and editing; Andrew P McMahon, Conceptualization, Resources, Formal analysis, Supervision, Funding acquisition, Methodology, Writing—original draft, Project administration, Writing—review and editing

### Author ORCIDs

Lori L O'Brien http://orcid.org/0000-0002-0741-181X
Alexander N Combes https://orcid.org/0000-0001-6008-8786
Odyssé Michos http://orcid.org/0000-0002-0002-4315
Ian M Smyth https://orcid.org/0000-0002-1727-7829

## Ethics

Animal experimentation: Studies were performed according to the recommendations in the Guide for the Care and Use of Laboratory Animals of the National Institutes of Health. All mouse handling and husbandry were performed according to guidelines issued by the Institutional Animal Care and Use Committees (IACUC) at the University of Southern California and after approval.(protocol #11893)

## Decision letter and Author response

Decision letter https://doi.org/10.7554/eLife.40392.031
Author response https://doi.org/10.7554/eLife.40392.032

## Additional files

### Supplementary files

• Supplementary file 1. RNA-seq from nephron progenitors. RNA-seq of E15.5 GFP +FAC sorted nephron progenitors from wild type and *Wnt11* mutants. Biological triplicates were performed for each genotype. Fold changes and RPKM are reported for each gene. Genes with RPKM <0 in wild type and mutant samples were removed for simplicity.
DOI: https://doi.org/10.7554/eLife.40392.026

• Supplementary file 2. RNA-seq from whole kidneys. RNA-seq of E15.5 whole kidneys from wild type and *Wnt11* mutant animals. Six biological replicates were performed for each genotype. Fold changes and RPKM are reported for each gene. Genes with RPKM <0 in wild type and mutant samples were removed for simplicity.
DOI: https://doi.org/10.7554/eLife.40392.027

• Transparent reporting form
DOI: https://doi.org/10.7554/eLife.40392.028

### Data availability

Sequencing data have been deposited in GEO under accession code GSE118334. All other data generated or analysed during this study are included in the manuscript and supporting files. Source data files have been provided where appropriate.

The following dataset was generated:

| Author(s) | Year | Dataset title | Dataset URL | Database and Identifier |
|---|---|---|---|---|
| O'Brien LL, Whitney PH, McMahon AP | 2018 | Differential gene expression between wild type and Wnt11 mutant embryonic kidneys | https://www.ncbi.nlm.nih.gov/geo/query/acc.cgi?acc=GSE118334 | Gene Expression Omnibus, GSE118334 |

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
