## [Decision Letter]

Thank you for submitting your article "Wnt11 signaling regulates the organization and actions of kidney nephron progenitors to determine nephron endowment" for consideration by *eLife*. Your article has been reviewed by three peer reviewers, including Jody Rosenblatt as the Reviewing Editor and Reviewer #1, and the evaluation has been overseen by Didier Stainier as the Senior Editor. The following individual involved in review of your submission has agreed to reveal their identity: Fernando Martín-Belmonte (Reviewer #3).

The reviewers have discussed the reviews with one another and the Reviewing Editor has drafted this decision to help you prepare a revised submission.

While I list the full reviews below, overall the reviewers were very pleased with the manuscript. Although several reviewers would like more mechanistic signaling detail, we felt that drilling down to this goes beyond the scope of the paper and that the cellular mechanistic findings are novel and an excellent addition to the body of knowledge on how Wnt11 signaling contributes to kidney development. The only points that we suggest are:

1) That the authors make the title clearer and more descriptive of the actual findings. Currently, we found it a bit vague and we think adding a bit of the mechanistic findings would recruit more interest and readership.

2) We found that there were too many figures, making the manuscript less punchy than it should be. Therefore, we suggest reducing some of the figures, described below. Please also check that the video can be played universally, too.

*Reviewer #1:*

The manuscript by O'Brien et al. reveals an interesting mechanism for how a polarity defect brought about by a Wnt11 mutation can disrupt kidney development by disrupting the proliferation/differentiation axis. They document with excellent quantitative evidence that the kidneys of Wnt11-/- mice are smaller due to premature differentiation of cells. This amplification of differentiation is due to disruption of interactions of the progenitors to their niche, which they show is due to alterations in cell polarity. I think that this is an interesting paper that shows very nicely how subcellular defects translate to whole tissue defects and think it will be of interest to many people in development. It is also unusual in that it shows that the cell-cell interactions override transcription, as there were no clear detectable transcriptional changes.

1) The title is not very clear or descriptive and should be changed to say more precisely what the conclusions say.

2) I could not get the one video to work. It would be nice to see more videos of this and determine if there is not also some defect in attachment of the dividing cell to its matrix that the authors don't discuss. I tried several ways to get the video to work, so make sure that their compressions allow access to several playback applications.

3) A schematic at the beginning of the paper showing the layout of the kidney cells divide and differentiate would be very helpful.

*Reviewer #2:*

O'Brien et al. show that nephron progenitors cannot maintain stable attachments to the ureteric bud tips in the absence of Wnt11, resulting in dispersion and premature depletion/differentiation of the progenitors, which could presumably explain nephron number reduction in the adult kidneys. The presented images are of high quality and the quantitative data are solid. Especially, time-lapse analysis convincingly demonstrates the unique movements of the mutant nephron progenitors, which can only be addressed with the authors' techniques.

Weakness: the authors fail to identify direct signaling defects downstream of Wnt11, despite the extensive analyses using multiple mutant mice and RNA-sequencing. It remains unclear which non-canonical pathway is affected, leading to the impaired polarity of the progenitors and their unstable attachments to the ureteric buds.

Nonetheless, the conclusion that Wnt11 functions through the non-canonical pathway in the kidney is novel and attractive. I believe that this manuscript is suitable for publication in *eLife*.

*Reviewer #3:*

In this manuscript, the authors present evidence for a novel action of ureteric branch tip-derived Wnt11 in progenitor cell organization and interactions within the nephrogenic niche, ultimately determining nephron endowment. They identified that this phenotype derived from the dispersion of nephron progenitors from their restricted niche in Wnt11 mutants, which intermix with interstitial progenitors. Indeed, they observe that nephron progenitors lose stable attachments to the ureteric branch tips in the absence of Wnt11, so they continuously detach and reattach from the ureteric cells.

Overall, the findings provided by the authors in this study are interesting, and most of the experiments of very high quality. It is essential to understand how the defects associated with the loss of function of essential molecules in renal development, such as Wnt11, are connected with specific cellular mechanisms of the progenitor cell population. I found this manuscript really well done, on a very high scientific level, and well written. In my opinion, only minor changes would need to be considered:

1) This manuscript, although very interesting and necessary, suffers from an explicit mechanistic characterization. The authors show how Wnt11 affects the polarization and attachment of the Six2 progenitors to the ureteric tip cells that generate this signal. However, we do not know at the molecular level how this response occurs, although the authors have presented us with some small details of this response, such as the location of integrins and the orientation of the Golgi (Figure 9). Therefore, the article format that is presented is probably excessive. A report format with 4-5 figures would probably be more appropriate. In general, Figures 1-4 are very descriptive, and in some cases provide redundant information. For instance Figure 2E and 3D show both kidney volume at different ages using different methods to estimated volume. Considering the length of the manuscript regarding the number of figures (9), the authors should consider reducing the first four figures to just two, and relocate some of the information to the supplementary section.

2) Similarly, I am not very convinced whether Figure 6 should be included among the central figures of the article. The conclusions derived from these data are relatively uninteresting for the scientific community. It is true that there is much redundancy in the Wnt receptors, and that therefore it is a recurrent problem when identifying the molecular mechanisms associated with a specific Wnt signaling (Wnt11 in this case). However, either the results of the mutant Ptk (Figure 6C) and that of the double mutant Fz2 / Fz7 (figure S3D) are characterized more in detail, or all this information is transferred to supplementary data.

3) Comparing kidney volume of younger (Figure 2E) vs. older kidneys (Figure 3D) younger kidneys appear to be bigger than older kidneys. Please explain.

4) Since *eLife* is not a journal specialized to nephrology topics, a sketch of the so-called "nephrogenic niche" would facilitate reading and understanding for non-renal-development experts.

5) Figure 4. Please introduce an example image where the regions Hull, Cap, Tip, Tree are labeled, and from where volume/length was calculated.

6) Please, explain abbreviations in the reading text

- Line 117: IRES (Internal Ribosomal Entry Site)

- Line 154: OPT (Optical Projection Tomography)

---

## [Author Response]

While I list the full reviews below, overall the reviewers were very pleased with the manuscript. Although several reviewers would like more mechanistic signaling detail, we felt that drilling down to this goes beyond the scope of the paper and that the cellular mechanistic findings are novel and an excellent addition to the body of knowledge on how Wnt11 signaling contributes to kidney development. The only points that we suggest are:1) That the authors make the title clearer and more descriptive of the actual findings. Currently, we found it a bit vague and we think adding a bit of the mechanistic findings would recruit more interest and readership.2) We found that there were too many figures, making the manuscript less punchy than it should be. Therefore, we suggest reducing some of the figures, described below. Please also check that the video can be played universally, too.

We would like to thank you and the other reviewers for the overall positive review of our submitted manuscript. We appreciate the comments and have revised as suggested. Namely, we have 1) given the manuscript a new, more descriptive title and 2) have shortened the number of figures.

Reviewer #1:The manuscript by O'Brien et al. reveals an interesting mechanism for how a polarity defect brought about by a Wnt11 mutation can disrupt kidney development by disrupting the proliferation/differentiation axis. They document with excellent quantitative evidence that the kidneys of Wnt11-/- mice are smaller due to premature differentiation of cells. This amplification of differentiation is due to disruption of interactions of the progenitors to their niche, which they show is due to alterations in cell polarity. I think that this is an interesting paper that shows very nicely how subcellular defects translate to whole tissue defects and think it will be of interest to many people in development. It is also unusual in that it shows that the cell-cell interactions override transcription, as there were no clear detectable transcriptional changes.1) The title is not very clear or descriptive and should be changed to say more precisely what the conclusions say.

We agree and have now changed the title to “Wnt11 directs nephron progenitor polarity and motile behavior ultimately determining nephron endowment”. We believe this more specifically states the mechanistic findings of Wnt11 actions in the kidney.

2) I could not get the one video to work. It would be nice to see more videos of this and determine if there is not also some defect in attachment of the dividing cell to its matrix that the authors don't discuss. I tried several ways to get the video to work, so make sure that their compressions allow access to several playback applications.

We apologize for the inability to play the video. We will upload a video file that has been viewed on several computer formats to ensure playback. In general, it appears that wild type nephron progenitors detach from the matrix in order to divide before reattaching. In the Wnt11 mutants, (an example can be seen in the supplementary video provided), cells already detached can undergo division. However, we have not captured sufficient numbers of dividing cells in our time lapse imaging to definitively say this. The minimal labeling approach limits the number of cells that are captured undergoing division.

3) A schematic at the beginning of the paper showing the layout of the kidney cells divide and differentiate would be very helpful.

We have included a schematic of the nephrogenic niche in Figure 1 to help orient readers.

Reviewer #3:[…] 1) This manuscript, although very interesting and necessary, suffers from an explicit mechanistic characterization. The authors show how Wnt11 affects the polarization and attachment of the Six2 progenitors to the ureteric tip cells that generate this signal. However, we do not know at the molecular level how this response occurs, although the authors have presented us with some small details of this response, such as the location of integrins and the orientation of the Golgi (Figure 9). Therefore, the article format that is presented is probably excessive. A report format with 4-5 figures would probably be more appropriate. In general, Figures 1-4 are very descriptive, and in some cases provide redundant information. For instance Figure 2E and 3D show both kidney volume at different ages using different methods to estimated volume. Considering the length of the manuscript regarding the number of figures (9), the authors should consider reducing the first four figures to just two, and relocate some of the information to the supplementary section.

Unfortunately, we were unable to determine a mechanism despite a strong effort; more complex genetic tools would be necessary to thoroughly address this. We have reduced the number of main figures from 9 to 7. Figure 6 has been moved to the supplement (Figure 4—figure supplement 1) and Figures 1 and 2 have been combined into Figure 1, with some of the data moving to the supplement (Figure 1—figure supplement 2). We feel this is a sufficient compromise. Although some information may appear redundant, measurements such as kidney size at different stages account for any changes in phenotype over time or phenomenon such as compensatory cellular hypertrophy which could occur.

*2) Similarly, I am not very convinced whether Figure 6 should be included among the central figures of the article. The conclusions derived from these data are relatively uninteresting for the scientific community. It is true that there is much redundancy in the Wnt receptors, and that therefore it is a recurrent problem when identifying the molecular mechanisms associated with a specific Wnt signaling (Wnt11 in this case). However, either the results of the mutant Ptk (Figure 6C) and that of the double mutant Fz2 / Fz7 (figure S3D) are characterized more in detail, or all this information is transferred to supplementary data*.

Figure 6 has been moved to the supplement (Figure 4—figure supplement 1).

3) Comparing kidney volume of younger (Figure 2E) vs. older kidneys (Figure 3D) younger kidneys appear to be bigger than older kidneys. Please explain.

The units were confusing between the two figures as they do not match. The units in Figure 2E (now Figure 1—figure supplement 2B) have been changed to mm^3^. (1 mm^3^ =1x10^9^ μm^3^).

4) Since eLife is not a journal specialized to nephrology topics, a sketch of the so-called "nephrogenic niche" would facilitate reading and understanding for non-renal-development experts.

A schematic of the nephrogenic niche has been added to Figure 1A.

5) Figure 4. Please introduce an example image where the regions Hull, Cap, Tip, Tree are labeled, and from where volume/length was calculated.

A schematic of the regions measured and what imaging was utilized for their analyses is included in what is now Figure 3A. We have removed the hull measurement as we felt it was not adding any additional information.

6) Please, explain abbreviations in the reading text- Line 117: IRES (Internal Ribosomal Entry Site)- Line 154: OPT (Optical Projection Tomography)

These have now been included in the text.